# SWE-bench Multimodal: Do AI Systems Generalize to Visual Software Domains?

**John Yang**[*1]  **Carlos E. Jimenez**[*2]  **Alex L. Zhang**[2]  **Kilian Lieret**[2]

**Joyce Yang**[3]  **Xindi Wu**[2]  **Ori Press**[4]  **Niklas Muennighoff**[1]  **Gabriel Synnaeve**[5]

**Karthik R. Narasimhan**[2]  **Diyi Yang**[1]  **Sida I. Wang**[5]  **Ofir Press**[2]

[1]Stanford University    [2]Princeton Language & Intelligence, Princeton University

[3]Cornell University    [4]Tübingen AI Center, University of Tübingen    [5]Meta AI

## Abstract

Autonomous systems for software engineering are now capable of fixing bugs and developing features. These systems are commonly evaluated on SWE-bench (Jimenez et al., 2024a), which assesses their ability to solve software issues from GitHub repositories. However, SWE-bench uses only Python repositories, with problem statements presented predominantly as text and lacking visual elements such as images. This limited coverage motivates our inquiry into how existing systems might perform on unrepresented software engineering domains (e.g., front-end, game development, DevOps), which use different programming languages and paradigms. Therefore, we propose SWE-bench Multimodal (SWE-bench M), to evaluate systems on their ability to fix bugs in visual, user-facing JavaScript software. SWE-bench M features 617 task instances collected from 17 JavaScript libraries used for web interface design, diagramming, data visualization, syntax highlighting, and interactive mapping. Each SWE-bench M task instance contains at least one image in its problem statement or unit tests. Our analysis finds that top-performing SWE-bench systems struggle with SWE-bench M, revealing limitations in visual problem-solving and cross-language generalization. Lastly, we show that SWE-agent's flexible language-agnostic features enable it to substantially outperform alternatives on SWE-bench M, resolving 12% of task instances compared to 6% for the next best system.

## 1 Introduction

Language models (LMs) are being increasingly deployed to assist software engineers (Bagalkote, 2024; Yepis & StackOverflow, 2024). As LMs gain in prominence, the research community has been expanding from building LM-based assistants that work at the code line or function level (Chen et al., 2021; Hendrycks et al., 2021) to building *autonomous systems* that can maintain and improve large codebases with hundreds of files and thousands of lines (Wang et al., 2024b; Xia et al., 2024; Yang et al., 2024; Zhang et al., 2024b). These systems provide LMs with tools and environments that let them engage in multi-step interactions to solve complex software development tasks.

SWE-bench (Jimenez et al., 2024a) is the most popular benchmark for evaluating the performance of these systems. Drawn from GitHub issues and pull requests, SWE-bench task instances capture a range of software bugs and verify solution behavior by executing unit tests. Since the introduction of SWE-bench in October 2023, state-of-the-art performance on SWE-bench Lite, the most commonly used subset of SWE-bench, has soared from 3% to 43% (Jimenez et al., 2024b).

However, with respect to the broader landscape of software engineering, SWE-bench reflects only a fraction of real-world applications. Its 17 repositories are predominantly written in Python. Task instance codebases tend to be structured similarly because every repository is a PyPI package. Though

---

* Equal contribution. Correspondence to `johnby@stanford.edu`, `carlosej@princeton.edu`.

Data, code, and leaderboard at swebench.com/multimodal.

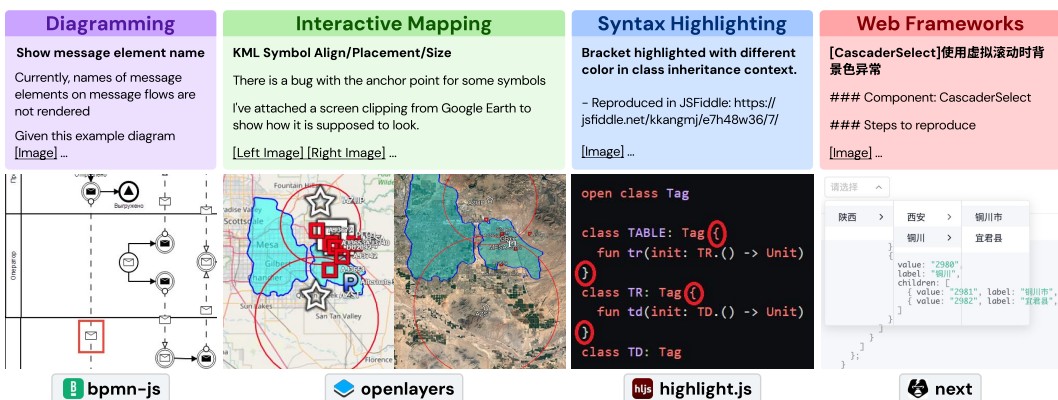

Figure 1: **Four task instances from SWE-bench Multimodal.** The 17 JavaScript repositories featured in SWE-bench M introduce new categories of software development challenges, such as the use cases shown above, along with performance profiling, art, and more. In addition to multimodal comprehension, SWE-bench M contains more multi-language issues for both programming (JavaScript, TypeScript, HTML, CSS), and natural (English, Chinese) languages.

SWE-bench features several commonly used backend and data science libraries, many other use cases are not represented. In addition, many domains of software development rely on visual assets, such as user interface design, gaming, virtual reality, and data visualizations (Foley & Van Dam, 1982; Shneiderman, 1998), but only 5.6% of SWE-bench tasks contain an image.

In light of these limitations, this paper poses the research question:

"Do AI Systems Generalize to Visual Software Domains?"

A popular sub-field capturing several of these unexplored dimensions is *front-end web development*, which focuses on building user-facing applications and websites with visual and interactive elements Si et al. (2024). Here, coding problems frequently contain both textual and visual components (Cherubini et al., 2007; Moody, 2009). As of 2023, JavaScript, the primary vehicle for front-end development, was the most popular programming language for the last decade (Daigle & GitHub, 2023). The JavaScript ecosystem encompasses a wide range of frameworks, such as Node.js, React, and Angular, each embodying unique architectural principles (Wittern et al., 2016).

In this work, we introduce SWE-bench Multimodal (SWE-bench M), a dataset of 619 task instances focused on visual JavaScript problems. Like SWE-bench, we derive instances from real-world issues on GitHub. Unlike SWE-bench, our collection targets JavaScript-based, user-facing applications, such as UI design systems, web app development, interactive mapping, and syntax highlighters, with four example instances shown in Figure 1. Furthermore, we filter exclusively for task instances that contain images or videos in their problem descriptions or testing scenarios. Finally, through verification by human experts, we find that for 83.5% of task instances, images are necessary for solving the corresponding issue.

We discover that **existing systems perform significantly worse on SWE-bench M than they do on SWE-bench**, due in large part to the challenges of visual problem solving and JavaScript's diverse development practices. The performance results on SWE-bench M appear to vary depending on the diverse types of images and challenges presented. Different visual elements, such as code snippets, website screenshots, and diagrams, seem to require distinct comprehension abilities. Furthermore, JavaScript's support for object oriented, functional, and procedural programming introduces substantial variance in how codebases are structured, which standardized solutions struggle with.

Our efforts to adapt existing systems highlight *generalizability* as a desirable but overlooked consideration for building LM systems. Current systems evaluated on SWE-bench rely heavily on Python-only parsers to perform fault localization steps independent of an LM (Örwall, 2024; Xia et al., 2024; Zhang et al., 2024b). These approaches either force system builders to re-engineer tools for other programming languages or fail entirely when similar tools are not available. Among our takeaways, we offer suggestions for building agent systems that operate efficiently in numerous programming languages and repositories with visual content.

Table 1: **Comparison of repository-level coding benchmarks.** We characterize benchmarks by their number of repositories (**# Repos**), programming languages (**Lang.**), whether they employ execution-based evaluation (**Exec.**), and include tasks with image content (**Images**). Programming languages for front-end apps and visual assets are novel attributes introduced by SWE-bench M.

| Dataset | # Repos | Lang. | Exec. | Images |
|---|---|---|---|---|
| RepoEval (Zhang et al., 2023b) | 14 | Python | ✗ | ✗ |
| RepoBench (Liu et al., 2023b) | > 10,000 | Python, Java | ✗ | ✗ |
| SWE-bench (Jimenez et al., 2024a) | 17 | Python | ✓ | ✗ |
| SWE-bench Multimodal | 17 | JavaScript | ✓ | ✓ |

## 2 SWE-BENCH MULTIMODAL

We first review SWE-bench's task formulation and limitations (Section 2.1) and describe how SWE-bench M builds upon this work. We then discuss SWE-bench M's data collection heuristics (Section 2.2). Finally, we thoroughly characterize SWE-bench M (Section 2.3), highlighting the novel challenges it poses to LM agent-based software development.

### 2.1 PRELIMINARIES

**Formulations.** SWE-bench (Jimenez et al., 2024a) has emerged as a popular LM agent benchmark. By drawing from real GitHub workflows, task instances reflect diverse, practical software challenges and require systems to ingest meaningful, long-form inputs. Repository-level coding also involves meticulous refactoring of interwoven modules, each with dependency trees spanning multiple source files. SWE-bench's collection strategy yields human-written unit tests that ensure robust evaluation.

SWE-bench consists of 2,294 task instances derived from pull requests (PRs) collected across 12 open source Python repositories. Each task instance corresponds to a PR and one or more resolved *issues*; issue(s) describe a bug or feature request, and the PR contains the corresponding solution code along with unit tests validating its correctness. The unit tests fail before the solution code is applied to the codebase but pass afterwards, also referred to as *fail-to-pass* (F2P) tests. Additional auxiliary *pass-to-pass* (P2P) tests verify that the codebase's existing functionality is maintained.

A task worker is shown the codebase and the issue, also called the *problem statement*. The worker must then modify the codebase to solve the problem. The proposed modification is run against both F2P and P2P tests to check if (1) the issue is fixed and (2) prior working behavior is not broken. If all unit tests pass, the task instance is considered resolved.

**Limitations.** Within the SWE-bench task formulation, several facets of software development remain unexplored. This work primarily investigates two such facets. First, SWE-bench task instances are predominantly *text only*, and there are no discussions of the implications of the interplay between images and videos with software development. For the 5.6% of SWE-bench task instances with an image, it is unclear what these images portray and whether they are necessary to solving the task. To fill this gap, we focus on task instances with visual elements and demonstrate the significance of multi-modal reasoning in software engineering evaluations. Second, SWE-bench is *comprised exclusively of Python repositories*. We show that a benchmark containing SWE-bench-style task instances from an alternative programming language, i.e., JavaScript, highlights previously unconsidered complexities (e.g., web development, asynchronous programming, DOM/state manipulation).

### 2.2 COLLECTION

JavaScript repositories have a high concentration of visual assets due to its popularity for full stack web development and browser manipulation. We summarize our modifications to SWE-bench's task collection pipeline to identify task instances with visual components. See Appendix B for details.

**1. Find user-facing JavaScript repositories.** Using GitHub's search feature, we look for JavaScript repositories with 5,000 or more stars and 500 or more pull requests. We then manually pick 17 repositories, filtering for user-facing libraries that have a visual aspect, such as mapping, plotting,

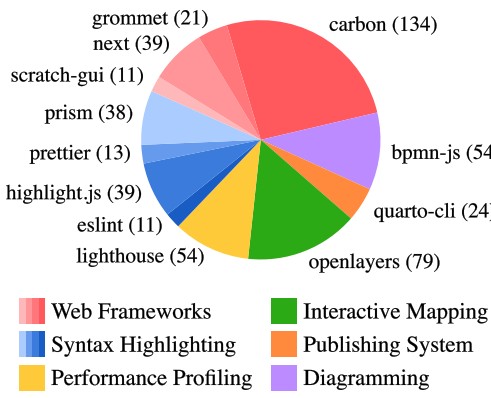

Figure 2: **SWE-bench M Test Split Distribution** of 517 task instances from 12 open source GitHub repositories written mainly in JavaScript.

Table 2: **Median values of different attributes of a SWE-bench M task instance.**

|  |  | Median |
|---|---|---|
| Issue Text | Length (Words) | 105 |
| Codebase | # Lines (non-test) | 549K |
|  | # Files (non-test) | 1799 |
| Gold Patch | # Lines edited | 27 |
|  | # Files edited | 2 |
|  | # Funcs edited | 3 |
| Tests | # Fail to Pass | 1 |
|  | # Pass to Pass | 5 |
| Images | Aspect Ratio | 5:3 |
|  | File Size (KB) | 42.95 |
|  | Resolution (Pixels) | 262K |

or syntax highlighting. We scrape 135k PRs from these selected repositories. Every repository's source code is at least 70% JavaScript or TypeScript code, with the remainder usually being HTML or Cascading Style Sheets (CSS). None of the selected repositories have Python code.

**2. Filter for issues that have visuals in their description or unit tests.** We first look for PRs with one or more issues. We then filter these [issue(s), PR] pairs down to ones with visual assets in either the issue or testing code. Specifically, we inspect the issue text and test patch for working hyperlinks that point to an image (e.g., `jpg`, `png`) or video (e.g., `gif`, `mov`). From 135k PRs, this new filtering criteria yields 1,478 candidate task instances.

**3. Set up the environment for each repository.** SWE-bench's Docker containers do not support JavaScript out of the box. Therefore, we first add foundational infrastructure, such as Node.js and Chrome, to support JavaScript execution, visual testing, and webpage rendering in-browser. Next, for each repository, we read its public contribution guidelines and then write tailored installation and testing scripts. This process requires several rounds of manual trial and error not only to validate the setup procedure, but also to overcome undocumented obstacles. Designing environments suited to run the majority of task instances took an average of ten hours of manual labor per repository. We are able to install the repository and execute tests for 679 of the 1478 task instances.

**4. Remove inconsistent tests.** During manual inspection, we found that a small percentage of tests exhibited inconsistency, i.e., given the same patch, a test's output (*pass* or *fail*) was inconsistent across multiple evaluation runs. This phenomenon was also reported in SWE-bench (Brown et al., 2024). To eliminate such cases, we run validation per task instance 10 times and remove tests with inconsistent results, reducing 679 candidates to 643 viable instances.

**5. Human validation.** To remove impossible task instances and perform dataset characterization that is difficult to conduct automatically, the authors manually inspect each task instance. We identify the kinds of image(s) provided and assess how necessary image(s) are to solving a task. We include the annotation procedure and queries in Appendix D. Annotators removed 24 impossible tasks for a final total of 619 task instances. We discuss takeaways from annotations in Section 2.3.

### 2.3 FEATURES

SWE-bench M is a dataset of 619 multimodal, JavaScript task instances collected from 17 open-source GitHub repositories. It contains a diverse set of libraries, including ones for data visualization, building diagrams, website UI components, displaying maps, and syntax highlighting. As shown in Figure 2, the test split consists of 517 task instances from 12 repositories. The development split contains 102 task instances from 5 repositories.

**Diversity of images.** Across all SWE-bench M task instances, there are 862 "problem statement" images, meaning the image hyperlink is in the issue text. These images feature a multitude of visual processing challenges. Our annotation procedure grouped these images into seven distinct

categories. Interpreting UI issues in *website screenshots* (401 images), such as malformed front-end layouts or accessibility issues, involves spotting design flaws and assessing spatial relationships between elements. Grounding visual cues from *code screenshots* (194) and *error messages* (54) to specific codebase entities can provide hints for bug localization. Images of *diagrams* (107), *art* (38), *maps* (35), and *data visualizations* (28) depict repository-specific challenges that address properly generating complex statistical, geographic, and creative content.

**Videos, actual/expected reference images.** SWE-bench M task instances often have multiple pictures (221 instances) or videos (70) to succinctly communicate helpful context and reproduction steps. For instance, `gifs` illustrate issues for 43 instances from `bpmn-js`, a diagramming toolkit. Unexpected behaviors are easily depicted by recording diagram interactions, drag-and-drop actions, and zooming in/out. 67 instances from `carbon`, a UI design system, contain "actual" and "expected" image pairs that highlight visual discrepancies and incorrect implementations.

**Visual testing.** A subset of 69 SWE-bench M task instances test submissions' functional correctness with pixel-level visual testing. This approach renders web pages and then compares screenshots pixel-by-pixel, identifying visual differences that functional unit tests cannot detect.

**Necessity of images.** To gauge the importance of problem statements' images, we pose two questions to human annotators. First, *does an image convey more information than the text in the image?* For instance, a syntax-highlighting bug screenshot would show several lines of code, but the text coloring is an important visual element. Annotators answer "Yes" to this question for 80% of problem statement images. Second, *is it feasible to solve a task instance without its associated image(s)?* Of 557 instances with image(s), 83.5% were considered crucial for resolving the corresponding task. When provided, images are essential for diagnosing software bugs, and the information they convey is necessarily visual. See Appendices D.2 and D.3 for details.

**Difficulty curve.** Human annotators estimated the time it would take for a developer to solve the task instances in the benchmark. This analysis shows that SWE-bench M features a range of difficulty levels, with tasks classified as follows: 13% take less than 15 minutes, 43% take 15 minutes to an hour, 38% take 1-4 hours, and 6% take more than 4 hours. Comparing these ratios to a similar study performed on SWE-bench (Chowdhury et al., 2024) shows that SWE-bench M tasks are longer on average overall. Reference solutions in SWE-bench M also edit more files, functions, and lines than in SWE-bench. See Appendices A.3 and D.4 for details.

## 3 EVALUATING ON SWE-BENCH M

We now discuss the challenge of generalization for existing open-source solutions for SWE-bench and describe how we have adapted these methods for evaluation on SWE-bench M (Section 3.1). Additionally, we provide details about our the setup of our experiments (Section 3.2).

### 3.1 DO EXISTING SYSTEMS GENERALIZE?

We attempted to run existing open-source systems that perform well on SWE-bench (Jimenez et al., 2024b) on SWE-bench M. However, when adapting these systems to SWE-bench M, we found that several solutions were so heavily tailored to Python and SWE-bench to the point that they were unusable for JavaScript repositories and SWE-bench M evaluation.

Our observations from exploring a number of existing solutions led us to identify *generalizability* as a desirable but overlooked property of automated software engineering systems. Specifically, do existing agent systems perform well on bugs that are not similar to the ones found in SWE-bench (i.e., non-Python or issues that include images)?

Though current LMs perform well in multiple programming languages (Cassano et al., 2022), a rigidly defined system can force LMs to follow a particular problem solving pipeline that restricts its capabilities. Although such workflow-oriented systems may work well for a specific type of repository or even programming language, they can fail when confronted with minor distribution shifts outside their original design parameters. Simply put, the shift from Python to JavaScript or the addition of the image modality exceeds the abilities of many existing approaches.

We attempt to adapt the top four open-source systems on the SWE-bench leaderboard (Jimenez et al., 2024b) to work on SWE-bench M tasks. For each system, we discuss below whether adaptations were feasible based on the preceding questions and the changes we made, if applicable.

**SWE-agent** (Yang et al., 2024) is a lightweight framework connecting an LM to an operating system and shell process. It integrates a text-based agent-computer interface (ACI) for LMs to edit files in addition to the ability to execute shell commands. The primary modifications necessary for SWE-bench was adapting the message processing and history functionality to work with images for multimodal LMs. To explore the effects of further adaptation, we evaluate three ACI configurations for SWE-agent: (1) SWE-agent Base, the original SWE-agent ACI, initially designed for evaluation on Python and SWE-bench tasks; (2) SWE-agent JS, which converts SWE-agent Base's editing interface to detect JavaScript edit errors, mirroring the linting feature used by SWE-agent for Python files; and (3) SWE-agent M, which extends the SWE-agent JS ACI to provide a simple web browser, screenshot, and image viewing capabilities, allowing models to visually reproduce image-based issues and verify changes or generated images.

**Agentless** (Xia et al., 2024) proposes a two stage localize-then-repair pipeline. A repository is first pre-processed with the Python `ast` module into a simplified overview of the repository's structure. Based on this syntax tree, the LM can then view files' content and select which files to fix. In the repair stage, based on the identified files, the LM then generates multiple candidate patches. The most commonly repeated solution among the candidates is its final submission.

The main adaptation needed is swapping out the Python `ast` module with a custom JavaScript parser written from scratch. Based on feedback from one of the authors, we also update Python-centric prompts with more JavaScript-oriented instructions. Without these changes, Agentless gets 0% resolved on SWE-bench M's development split. We denote our adapted version Agentless JS.

**AutoCodeRover** (Zhang et al., 2024b) employs a two-phase approach similar to Agentless. The first phase involves a customized retrieval process where an LM performs multiple iterative searches. This is followed by a patch generation and testing phase, where an LM, provided with the search history and results, generates and refines a code patch using results from executing tests.

We found that most tools AutoCodeRover provides to the agent rely heavily on Python-specific program analysis features and even prior knowledge of the particular repositories included in SWE-bench. We ultimately chose not to benchmark AutoCodeRover since its specialized tools would require extensive redesign for SWE-bench M, likely resulting in a fundamentally different system.

**Moatless** (Örwall, 2024) subscribes to the localize-then-repair workflow, but does not use an LM for localization. Instead, code files are first converted in abstract syntax trees (AST). These ASTs are then aggregated into a single, searchable code graph represented as a Faiss index (Johnson et al., 2019). The LM then queries this index to generate a repair.

We worked off the author's in-progress implementation for generating ASTs from JavaScript and TypeScript files[1]. Although the code graph representation is language agnostic, the input AST representation needed to generate the Faiss index is heavily based on Python's object oriented design. As reflected by the magnitude of the pull request's changes, writing an equivalent parser for JavaScript that (1) reflects the code and (2) subscribes to the index's input format is non-trivial due to usage of functional and declarative programs being more commonplace. We do not benchmark Moatless.

**RAG.** Jimenez et al. (2024a) proposed a retrieval augmented baseline using BM25 (Robertson & Zaragoza, 2009) from which we inherit the document formatting, retrieval method, and prompt structure. We adapt the prompt template used in Jimenez et al. (2024a) to use a JavaScript example for patch generation instead of Python. We also include a new section in the prompt for inserting reproduction code collected from links in the problem statement, as described in B.3.

## 3.2 EXPERIMENT SETUP

**Models.** Resolving SWE-bench M issues with existing systems requires handling very long contexts, processing images and text simultaneously, and producing sophisticated structured outputs. Given these constraints, we focus all of our evaluations on GPT-4o (`gpt-4o-2024-08-06`) (OpenAI, 2024) and Claude 3.5 Sonnet (`claude-3-5-sonnet-20240620`) (Anthropic, 2024), the two

---

[1]https://github.com/aorwall/moatless-tools/pull/34

Table 3: **Performance comparison of various baselines on SWE-bench M.** The table shows results for different software development agent frameworks, including SWE-agent (with multimodal and JavaScript-specific variations) and a retrieval augmented generation (RAG) approach. Each system's success rate (% Resolved) and average cost ($ Avg. Cost) per task are reported.

| System | Model | % Resolved | $ Avg. Cost |
|---|---|---|---|
| SWE-agent M | GPT-4o | **12.2** | 2.94 |
| | Claude 3.5 Sonnet | 11.4 | 3.11 |
| SWE-agent JS | GPT-4o | 9.2 | 0.99 |
| | Claude 3.5 Sonnet | 12.0 | 3.11 |
| SWE-agent Base | GPT-4o | 12.0 | 2.07 |
| | Claude 3.5 Sonnet | **12.2** | 1.52 |
| Agentless JS | GPT-4o | 3.1 | 0.38 |
| | Claude 3.5 Sonnet | 6.2 | 0.42 |
| RAG | GPT-4o | 6.0 | 0.17 |
| | Claude 3.5 Sonnet | 5.0 | 0.15 |

most well-supported mulitmodal LMs for long-context RAG and agent systems. Though alternative models continue to improve, they currently lack the combination of long-context handling, multimodal processing, and structured prediction abilities required to work with existing autonomous software engineering systems.

**Baselines.** Based on our Section 3.1 findings, we benchmark the performance of five systems: Retrieval Augmented Generation (RAG), SWE-agent (Base, JS, M), and Agentless JS. Each of our baseline systems were adapted and developed using the development split of SWE-bench M. We describe more about how the final test configurations, including hyperparameters, were selected in Appendices C.1 and C.2.

**Metrics.** We report two evaluation metrics: (1) **% Resolved**, the main performance metric, which represents the proportion of successfully resolved task instances and (2) **Avg. $ Cost**, the mean per-instance inference cost incurred from running the baseline, as used in Yang et al. (2024).

## 4 RESULTS

We compare the performance of each baseline system in Table 3. While overall performance on SWE-bench M is relatively low, we observe a substantial performance gap between the interactive SWE-agent systems (11.5 % resolved on average) and the Agentless and RAG baselines, which achieve 3.9 and 5.5 % resolved on average respectively.

Across all SWE-agent configurations, we observe similar absolute performance, suggesting that the JavaScript-specific customizations in SWE-agent JS and SWE-agent M had minimal impact on performance. Additionally, we also observe minimal performance differences when swapping the underlying LM (GPT-4o or Claude 3.5 Sonnet) within a system. However, a study of SWE-agent ablations on the development set reveals that the added multimodal tooling can improve agent performance in some cases, though the overall picture remains ambiguous.

An analysis of solution rates segmented by the date each task instance was originally solved, is performed in Appendix C.3. It finds no evidence for a test set advantage from solutions leaking into the LM training sets: we show for example, that SWE-agent M using GPT-4o performs better on task instances based on issues that were resolved *after* its knowledge cutoff date.

### 4.1 ANALYSIS

We present insights into how images influence system performance, evaluate the impact of multimodal actions in SWE-agent M, and explore the generalization of different approaches to automated software engineering. All analyses are conducted on the development set of SWE-bench M.

**Resolving SWE-bench M issues requires improved visual understanding.** Table 4 compares the performance of the RAG and SWE-agent JS systems, with and without images. The results

reveal a significant performance decrease when models lack access to visual information. Table 5 analyzes performance based on the two key questions from our human validation: "Do the images contain more information than just text?" and "Are the images necessary to solve this task?" (For detail, see Appendices D.2 and D.3, respectively.) As expected, system performance generally improves when images are provided with the problem statement. In addition, for instances where annotators deemed images necessary to solve the task, performance plummets without visual input. Performance changes vary based on image content. When images contain primarily text, SWE-agent JS maintains 23.1% performance with or without images. Larger performance drops occur when images contain non-textual elements. For non-textual visuals, performance drops from 13.0% to 8.7% without images for SWE-agent JS. Visual elements, especially those that are non-textual, appear to provide additional context that substantially improves system performance.

Table 4: **Performance of RAG and SWE-agent JS with (W/) and without (W/o) Images.** The consistent improvement across both models and systems emphasizes the critical role of multimodal inputs, particularly for complex tasks requiring visual reasoning.

| System | Model | W/ Images | W/o Images |
|---|---|---|---|
| SWE-agent JS | GPT-4o | **11.0** | 8.0 |
| | Claude Sonnet 3.5 | **16.0** | 13.0 |
| RAG | GPT-4o | **10.0** | 8.0 |
| | Claude Sonnet 3.5 | **14.1** | 11.2 |

Table 5: **System performance segmented by annotator response**. We show performance for SWE-agent JS and RAG using Claude 3.5 Sonnet broken down by response to human validation questions detailed in Appendix D.2 and D.3. Each performance value corresponds to the performance of the configuration (e.g. SWE-agent JS with images) on the subset of instances corresponding to the annotation class for each question. The questions in the tables have been simplified from the original for readability.

| **Does the image contain more information than just text?** | | | | **Are the images necessary to solve this task?** | | | |
|---|---|---|---|---|---|---|---|
| System | With Images? | Yes | No | System | With Images? | Yes | No |
| SWE-agent JS | ✔ | 13.0 | 23.1 | SWE-agent JS | ✔ | 17.6 | 11.1 |
| | ✘ | 8.7 | 23.1 | | ✘ | 8.8 | 22.2 |
| RAG | ✔ | 11.6 | 13.8 | RAG | ✔ | 12.4 | 11.9 |
| | ✘ | 6.1 | 16.2 | | ✘ | 9.1 | 8.1 |

**Localization modules are overly engineered for Python.** As discussed in Section 3.1, except for SWE-agent, the systems that we study (Agentless, Moatless, and AutoCodeRover) impose fixed, procedural problem-solving workflows. Every system starts with a bug localization step that relies on abstract syntax tree (AST) parsing libraries to identify programmatic symbols.

These Python-specific modules do not work for other languages. Overcoming this design is contingent upon either finding or creating a similarly reliable tool. For Moatless and AutoCodeRover, no such replacement tool exists; reimplementation was prohibitively laborious and would have resulted in a system with little resemblance to the original. For Agentless, replacing invocations of the Python `ast` package with the JavaScript `tree-sitter` library still yielded a 0% resolve rate on the development split; this is due to Python-centric assumptions the localization module makes about the AST representations. Therefore, we wrote a custom JavaScript parser from scratch compatible with the Agentless logic, which took 15 hours of labor.

However, because the original design of Agentless is influenced by Python, fundamental differences between Python and JavaScript's design patterns (e.g., multiple programming styles, object prototypes, arrow functions) lead to many failure modes during localization, resulting in Agentless JS's low performance (4.6% in Table 3). For instance, for `grommet__grommet-6749`, the localization module, designed based on Python's *imperative* and object-oriented style, does not recognize `Tab`,

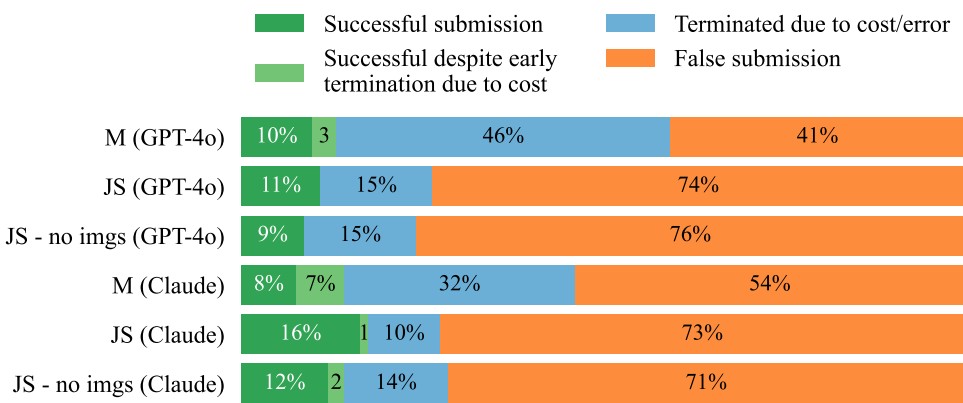

Figure 3: **SWE-agent performance across configurations.** The frequency of outcomes by success and exit status of SWE-agent under different configurations on the development set. Configurations include multimodal (M), JavaScript-specific (JS), and no images, for both GPT-4o and Claude 3.5 Sonnet models. A similar analysis by repository is shown in Figure 8.

the buggy component, because it was defined *declaratively* as an arrow function. In the subsequent repair phase, Agentless JS writes a completely new `Tab` imperatively, while the reference solution makes a minor change to its existing declarative implementation. We also found that given the raw code, GPT-4o correctly identified `Tab` as the buggy component.

We calculate the F1 Score for file localization on the development split with Claude Sonnet 3.5 as the base LM. Agentless JS achieves a score of $0.142$, while SWE-agent gets $0.367$. Flaws with language-specific design are only amplified in SWE-bench M. Unlike SWE-bench solutions, which edit only Python files, 28% of reference solutions in SWE-bench M edit multiple files types (e.g., TypeScript, HTML, CSS, Lua). Adapting current solutions requires writing unique AST generator and navigator functions to pre-process code files for many languages.

From these observations, *we hypothesize that generalizable LM-based systems for software engineering will emphasize interaction, not problem solving.* Instead of inserting LMs at specific points of a workflow, our results show the superiority of an LM-first approach that focuses on building tools that enhance an LM's ability to navigate and manipulate an environment. SWE-agent's superior performance and Agentless's localization struggles suggest that *the burden of problem solving should remain on the LM instead of being offloaded to manually engineered pipelines.*

**Multimodal tools increase task complexity but can improve agent performance.** To evaluate the impact of the presence of images on agent behavior, we conduct an analysis of the SWE-agent trajectories generated while resolving the development set. We run SWE-agent in three different configurations: SWE-agent M, SWE-agent JS, and SWE-agent JS without image inputs. The web-specific tools introduced in SWE-agent M lead to approximately 20% of actions being dedicated to building and taking screenshots of websites, though this proportion varies significantly across repositories (see Figures 6 and 7). SWE-agent M with GPT-4o builds websites and takes screenshots for 38.3% of the instances. For these instances, an average of 7.5 screenshots are taken, showing that the multimodal tooling is used as part of an iterative problem-solving process for some instances.

However, reproducing issues and verifying solutions by building and taking screenshots of websites makes the task for SWE-agent M more complex than directly performing and submitting edits as in SWE-agent JS. This is reflected in an almost threefold increase of attempts that are terminated because they exceed their cost limit (see Figure 3). The web-specific tools lead to degraded performance with Claude 3.5 Sonnet, while GPT-4o improves, even though some of its successes are from runs that are terminated prematurely due to costs.

Another key figure for agent performance is the *fraction of correct submissions*, as incorrect submissions increase manual reviewing workload. To increase this fraction, submissions from SWE-agent runs that were terminated prematurely due to cost can be excluded, as they have a lower probability

to be correct. In this setup, *adding multimodal tools almost doubles the fraction of* correct *submissions with GPT-4o* to 19.6% compared to 10.4% without image inputs. This result is not observed in the Claude 3.5 Sonnet-based SWE-agent.

Together, these findings show that LM agents can use web-based multimodal tools to reproduce issues and verify solutions, in some cases increasing success rates or solution correctness. However, using these tools to their full potential requires complex workflows that are still challenging for agents and result in an ambiguous record overall.

## 5 RELATED WORK

**Multimodal Code Benchmarks.** Code generation, the task of synthesizing code from natural language, has served as a long-standing measure of LM performance (Austin et al., 2021; Chen et al., 2021; Hendrycks et al., 2021). As performance on these benchmarks has plateaued—Claude Sonnet 3.5 achieves 92% on HumanEval—subsequent works have extended this task along different dimensions to increase difficulty, such as more robust evaluation (Liu et al., 2023a; Jain et al., 2024), multilinguality (Cassano et al., 2022; Wang et al., 2023a; Zheng et al., 2024), open-domain generation (Wang et al., 2023b; Zhuo et al., 2024), data science (Lai et al., 2022; Yin et al., 2022; Cao et al., 2024), understanding execution traces (Gu et al., 2024; Muennighoff et al., 2024), repository comprehension (Liu et al., 2023b; Zhang et al., 2023b), cybersecurity (Yang et al., 2023b; Zhang et al., 2024a; Shao et al., 2024; Abramovich et al., 2024), and efficiency (Huang et al., 2024; Liu et al., 2024; Waghjale et al., 2024). SWE-bench M represents a merger of two promising directions: software engineering (Jimenez et al., 2024a) and multimodal code generation (Li et al., 2024; Si et al., 2024; Wu et al., 2024; Nishina & Matsui, 2024). By drawing on real-world problems from GitHub, SWE-bench M overcomes the synthetic and short-form limitations inherent in code generation task formulation. In addition, SWE-bench M represents a significant multimodal upgrade to SWE-bench, which has already become a popular benchmark for LM evaluation.

**LM Agents for Web and Code.** Prior to the proliferation of LMs, earlier works explored translating between user interfaces and front-end code (HTML/CSS/JavaScript) (Beltramelli, 2018; Robinson, 2019). More recently, LM agents (Yao et al., 2023b; Sumers et al., 2024) have been increasingly employed and evaluated on various tasks (Wang et al., 2024a; Xie et al., 2024). Two popular use cases, web or application navigation (Hong et al., 2023; Soselia et al., 2023; Yan et al., 2023; Yao et al., 2023a; Zhang et al., 2023a; Koh et al., 2024; Putta et al., 2024; Rawles et al., 2024; Press et al., 2024; Yoran et al., 2024) and software engineering (Yang et al., 2023a; 2024; Zhang et al., 2024b; Xia et al., 2024), have typically been studied separately. To the best of our knowledge, SWE-bench M is the first benchmark that meaningfully couples these two tasks. Though prior approaches have provided web browsing tools to programming agents (Wang et al., 2024b), such tools usually represent webpages in text form and were not designed with a clear downstream objective. SWE-bench M demonstrates how agents can iterate meaningfully between code updates and their effects rendered as images or in a browser.

## 6 CONCLUSION

This work introduces SWE-bench Multimodal (SWE-bench M), the first benchmark to evaluate coding agents on real-world software engineering tasks involving visual elements. SWE-bench M contains 619 task instances from 17 user-facing JavaScript repositories, including ones for web user interface design, data visualization, art and mapping. Our analysis reveals that SWE-bench M contains diverse visual challenges and increases task complexity compared to SWE-bench. Furthermore, existing systems perform poorly on it, with the top resolve rate reaching only 12.2%; SWE-bench M presents repository-level programming tasks that are challenging even for state-of-the-art systems built on top of the strongest LMs. Incorporating multimodality in SWE-bench M not only expands the coverage of exciting, practical challenges in software engineering, but it also encourages practitioners to develop more general-purpose, language-agnostic solutions that do not overfit to SWE-bench or Python repositories.

ACKNOWLEDGEMENTS

We thank the authors of the Agentless and Moatless systems, Steven Xia and Albert Örwall, for responding to our questions and verifying implementation details during our attempts to modify these approaches to work on SWE-bench M.

We thank Open Philanthropy, Oracle and the National Science Foundation (Grant No. 2239363) for providing funding for this work. Any opinions, findings, conclusions, or recommendations expressed in this material are those of the author(s) and do not necessarily reflect the views of the National Science Foundation. Ori Press thanks the International Max Planck Research School for Intelligent Systems (IMPRS- IS) for its support.

All experiments were conducted by the authors at Stanford University and Princeton University on Princeton University's servers. Meta affiliated authors acted in an advisory role.

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

APPENDIX

In the appendix, we provide additional details about the collection process for curating SWE-bench M, more characterizations of the SWE-bench M dataset, supplementary experiments and ablations of LM agent performance on SWE-bench M, limitations and more.

## A  DATASET

### A.1  DEVELOPMENT SPLIT

As shown in Figure 4, the SWE-bench M development split consists of 100 tasks instances from 5 open source repositories largely written in the JavaScript family of programming languages (e.g. `js`, `jsx`, `ts`, `tsx`). Repositories were chosen for the development split based on two criteria. First, the number of task instances is about one-fifth the size of the test split. Second, each repository has an "equivalent" test repository, meaning for each repository in the development set, there is a repository with a similar purpose in the test set. This parallelism is meant to make solutions crafted on the development set more transferable to the test set.

We have observed that, in practice, practitioners developing solutions to the SWE-bench task have often iterated on task instances from the *test* split of the dataset. For those interested in working on SWE-bench M, we encourage practitioners to follow the preferable practice of iterating on the *development* split, and then only evaluate on the test split once the approach is finalized.

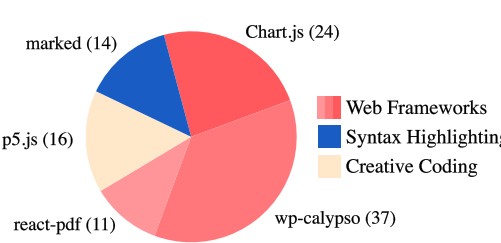

Figure 4: Distribution of SWE-bench M development set tasks (in parenthesis) across 5 open source GitHub repositories.

Table 6: Median values of different attributes of a SWE-bench M task instance.

|  |  | Median |
| --- | --- | --- |
| Issue Text | Length (Words) | 145 |
| Codebase | # Lines (non-test) | 228K |
|  | # Files (non-test) | 1324 |
| Gold Patch | # Lines edited | 35 |
|  | # Files edited | 2 |
|  | # Funcs edited | 3 |
| Tests | # Fail to Pass | 2 |
|  | # Pass to Pass | 7 |

As reflected in Table 2 compared to Table 6, a task instance from the development split has a slightly longer issue text, but also a smaller codebase in terms of number of lines and files. The median number of files (2) and functions (3) edited is identical. Development split task instances change slightly more lines, and have generally have somewhat more Fail-to-Pass and Pass-to-Pass tests.

### A.2  ADDITIONAL CHARACTERIZATIONS

We provide further characterizations about the SWE-bench M dataset. In this section, we include both extended discussions of filtering criteria and features discussed in Sections 2.2 and 2.3 as well as additional, noteworthy details not mentioned in the main paper.

**Visual Testing.** A total of 69 task instances from `Chart.js`, a data visualization framework, and `openlayers`, an interactive mapping library, verify functional correctness with pixel-level visual testing. Across this subset of instances, there are 273 reference screenshots for visual testing.

Within the JavaScript development ecosystem, visual testing like this is commonplace, with Puppeteer [2] and Pixelmatch [3] being two popular libraries. Such libraries are necessary because they provide several sensible advantages over a simple binary comparison of two images to check if

---

[2]https://github.com/puppeteer/puppeteer
[3]https://github.com/mapbox/pixelmatch

they're identical. Generally, the overarching reason is that a direct `diff` is too strict and doesn't account for the discrepancies of how different browser render user interfaces and webpages.

There are several additional benefits. First, such libraries provide the ability to specify tolerances for minor pixel level differences that might arise from anti-aliasing or sub-pixel rendering (Kotsarenko, 2018). Second, visual testing libraries allow for targeting certain components of a webpage. Specifically, Puppeteer has tools for taking screenshots of specified sub-parts of a page, and Pixelmatch allows for region-specific comparisons of images. This use case is particularly important for selectively ignoring dynamic website content, such as timestamps or ads, that is irrelevant to the code.

**Task instances by year.** In the following Table 7, we show the number of task instances for each repository across different years. Task instances are distributed fairly uniformly, with the majority of tasks being from 2019 to 2022. Some repositories have a higher distribution of older task instances, such as `lighthouse` and `wp-calypso`.

Table 7: Number of task instances per year for each repository represented in SWE-bench M.

| **Repository** | ≤ 2018 | 2019 | 2020 | 2021 | 2022 | 2023 | 2024 |
|---|---|---|---|---|---|---|---|
| bpmn-js | 0 | 22 | 6 | 4 | 18 | 4 | 0 |
| carbon | 0 | 36 | 30 | 30 | 24 | 12 | 2 |
| eslint | 4 | 3 | 0 | 3 | 0 | 1 | 0 |
| grommet | 3 | 1 | 0 | 1 | 11 | 5 | 0 |
| highlight.js | 0 | 0 | 14 | 21 | 4 | 0 | 0 |
| lighthouse | 25 | 12 | 7 | 3 | 3 | 4 | 0 |
| next | 3 | 9 | 11 | 10 | 4 | 0 | 2 |
| openlayers | 3 | 4 | 12 | 16 | 22 | 14 | 8 |
| prettier | 3 | 1 | 3 | 1 | 1 | 3 | 1 |
| prism | 5 | 6 | 8 | 14 | 5 | 0 | 0 |
| quarto-cli | 0 | 0 | 0 | 0 | 8 | 16 | 0 |
| scratch-gui | 7 | 2 | 0 | 0 | 1 | 1 | 0 |
| Chart.js | 0 | 0 | 2 | 17 | 3 | 2 | 0 |
| marked | 2 | 2 | 6 | 1 | 2 | 1 | 0 |
| p5.js | 1 | 4 | 1 | 1 | 6 | 3 | 0 |
| react-pdf | 1 | 1 | 0 | 8 | 0 | 1 | 0 |
| wp-calypso | 26 | 11 | 0 | 0 | 0 | 0 | 0 |
| Total | 83 | 114 | 100 | 130 | 112 | 67 | 13 |

**Asset collection.** For each task instance, we download all image, video, and reproduction code assets referenced by hyperlinks in the issue. This step, not done in SWE-bench, ensures that problem statements are reproducible in the future even if these hyperlinks expire.

**Multilingual problem statements.** 55 task instances have languages other than English in the problem statement text or images, with Mandarin Chinese being the second most frequently occurring language (38 task instances). The `next` repository, a design component library based on React, contributes the most of these instances, with 26 task instances. This is in large part due to the fact that the repository's owner is the Alibaba Design team. Although not explicitly emphasized in SWE-bench M's collection process, we find that collecting from repositories owned by maintainers whose primary language is not English is a fruitful source of multilingual SWE-bench style task instances.

**References solutions edit multiple file types.** In SWE-bench, all reference solutions edit exclusively Python files. In SWE-bench M, 174 task instances have references solutions that modify two or more file types. We did not consider non-code files such as text, Markdown, or images files when calculating the counts. We list the frequencies for the different sets of file types edited by reference solutions in Table 8.

## A.3 ADDITIONAL ANALYSES

We provide further details and breakdowns of the SWE-bench M dataset, such as repository specific statistics and additional information from scraping task instance information on GitHub.

Table 8: Counts for the number of task instances with reference solutions that edit a specific set of file types. 28% of SWE-bench M task instances modify two or more files types.

| File Types Edited | Count | File Types Edited | Count |
|---|---|---|---|
| js | 400 | js, scss, jsx | 8 |
| js, scss | 52 | lua | 8 |
| jsx | 19 | js, hbs, scss | 6 |
| ts, js | 18 | html, js, css | 4 |
| js, jsx | 16 | scss, jsx | 4 |
| html, js | 14 | js, lock | 3 |
| ts | 10 | js, css | 3 |

**Repository Statistics.** Following Tables 2 and 6, we show the same median statistics for each repository in Table 9. The discrepancies in the issue text length, codebase size, and gold patch changes can be directly attributed to the codebases themselves - relative to the original online sources for the code and issue text, the SWE-bench M collection process does not affect these values at all. On the other hand, the differences in the number of tests per repository, particularly Pass-to-Pass tests, is an outcome of how the testing specifications and test log parsing are manually specified. Two factors lead to these differences. First, for some repositories, the entire testing suite is run (e.g., `highlight.js`, `p5.js`, `quarto-cli`), while for the majority of repositories, only specific test files, usually derived from file paths modified by the test patch, are run (e.g., `carbon`, `openlayers`, `wp-calypso`). Consequently, for the latter set of repositories, the number of Pass-to-Pass tests is usually few to none. Second, the granularity of test logging is also different across codebases. The primary distinction is that while some test logs show Pass/Fail on a per-file basis, other test logs will show Pass/Fail on a per-test case basis where one files has multiple test cases.

Table 9: **Median values for metrics about each task instance,** mirroring the statistics shown in Tables 2 and 6. Repositories from the test set are listed above the separator, while development set repositories are listed below. "Fail to Pass" is abbreviated as "F2P." "Pass to Pass" is "P2P."

| Repository | Issue Text Length | Codebase Lines | Codebase Files | Gold Patch Lines | Gold Patch Files | Gold Patch Funcs | Tests F2P | Tests P2P |
|---|---|---|---|---|---|---|---|---|
| GoogleChrome/lighthouse | 94 | 1076K | 834 | 29 | 2 | 3 | 1 | 1 |
| PrismJS/prism | 79 | 184K | 2,721 | 12 | 2 | 2 | 1 | 9 |
| alibaba-fusion/next | 14 | 200K | 1,731 | 57 | 5 | 5 | 3 | 25 |
| bpmn-io/bpmn-js | 92 | 99K | 593 | 15 | 1 | 2 | 1 | 0 |
| carbon-design-system/carbon | 116 | 822K | 4,250 | 45 | 3 | 4 | 1 | 33 |
| eslint/eslint | 324 | 398K | 1,567 | 23 | 1 | 2 | 2 | 0 |
| grommet/grommet | 105 | 612K | 1,186 | 8 | 1 | 1 | 1 | 77 |
| highlightjs/highlight.js | 139 | 105K | 1,476 | 16 | 2 | 2 | 2 | 1,641 |
| openlayers/openlayers | 128 | 544K | 1,761 | 32 | 2 | 3 | 1 | 0 |
| prettier/prettier | 105 | 348K | 6,855 | 48 | 2 | 5 | 1 | 0 |
| quarto-dev/quarto-cli | 202 | 1399K | 3,639 | 12 | 2 | 2 | 1 | 434 |
| scratchfoundation/scratch-gui | 148 | 100K | 514 | 562 | 12 | 9 | 1 | 0 |
| Automattic/wp-calypso | 121 | 562K | 7,990 | 76 | 3 | 5 | 2 | 11 |
| chartjs/Chart.js | 148 | 154K | 1,376 | 18 | 1 | 2 | 1 | 0 |
| diegomura/react-pdf | 88 | 188K | 806 | 39 | 2 | 2 | 1 | 210 |
| markedjs/marked | 89 | 38K | 283 | 11 | 1 | 1 | 1 | 0 |
| processing/p5.js | 254 | 436K | 1,009 | 30 | 2 | 4 | 12 | 2,373 |

As shown in Figure 5, we include plots of the cumulative distribution functions for the statistics presented in Table 9 across all task instances from both the development and test splits. We also do the same calculations for the SWE-bench dataset (2519 task instances total) and overlay the CDFs for each statistic. The frequencies are normalized to a range of zero to one.

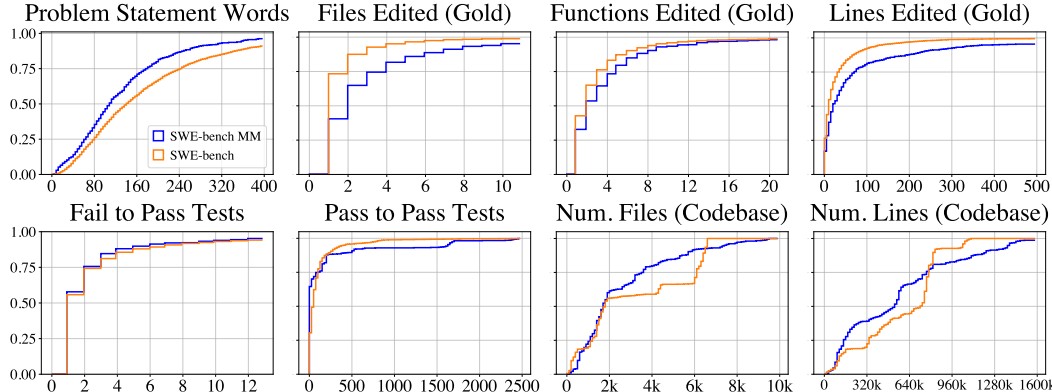

Figure 5: Normalized cumulative distribution functions for different task instance statistics. We compare these statistics between SWE-bench M (blue) and SWE-bench (orange).

While the median number of files and functions is low, with a majority of changes being local and limited to a few lines, there is a long tail of tasks that edit multiple files and functions. The distribution of edits is unlike SWE-bench's. In SWE-bench, 83% of task instances edit one file, and 65% edit one function. In SWE-bench M, just 40% of task instances change one file, and 32.5% change one function. On average, the changes by reference solutions in SWE-bench M are larger than those in SWE-bench, with multi-file edits being more commonplace.

The average SWE-bench M codebase is somewhat smaller than a SWE-bench repository. The median SWE-bench M codebase has 200k fewer lines of code than its SWE-bench counterpart (535k compared to 734k). However, in the long tail, SWE-bench M has codebases that are larger in terms of both number of files and lines.

Testing compared to SWE-bench is quite similar. For SWE-bench, 80% of task instances have two or fewer fail to pass tests compared to 75% for SWE-bench M. Finally, another noticeable trend is that SWE-bench M issues tend to be shorter than SWE-bench problem statements. This trend can be attributed to the inclusion of images as a complementary form of communication.

**GitHub Issue Tags.** For every issue collected in the SWE-bench M dataset, we check what labels, if any, were assigned to each issue. These human-annotated labels provide further insight into the kinds of fixes and contributions that a SWE-bench M task instance makes to the codebase. Table 10 summarizes the categories of tags along with the counts of individual tags.

Table 10: **Categories of** 1018 **tags associated with** 535 **issues associated with one of the** 517 **task instances** from the test split of SWE-bench M (499 task instances have 1 issue, and 18 have 2).

| Category | Count | Examples |
|---|---|---|
| Bug | 267 | "bug" (207); "type: bug" (56) |
| Feature | 79 | "type: enhancement" (44); "enhancement" (19); "feature request" (10); "feature" (6); |
| Other | 672 | "a11y" (40); "language" (37); "help welcome" (35); "language-definitions" (32); "role: dev" (32); "good first issue" (28); "package: react" (21); "package: @carbon/react" (14); modeling (13); |

While bug fixes are the most popular type of tag in terms of raw frequency, they only constitute a quarter of all tags. In addition to features, there are a variety of more repository specific bug fixes that highlight the diversity of the types of fixes corresponding to SWE-bench M task instances. For example, the "a11y" tag, shorthand for "accessibility", is used in both the `carbon` and `bpmn-js` repositories. Issues with this tag usually describe how the accessibility features of a UI component from the JavaScript library is either missing or defective. The "language" and "language-definitions" tags are used by syntax highlighting libraries including `highlight.js` and `prism`. Issues with this tag refer to bugs where language-specific symbols are not being highlighted correctly in a code

editor. The "modeling" tag, used exclusively in the `bpmn-js` repository, refers to unexpected behavior the user encounters when manipulating `bpmn-js` diagrams. Every SWE-bench M task instance associated with an issue that has this tag contains either a video or annotated picture that captures the phenomenon via screen recording and/or describes what the expected action should be. Such problem statement require not only visual reasoning acumen, but also the ability to ground accompanying issue description text in the visual elements.

Table 11: **Categories of** 186 **tags associated with** 109 **issues associated with one of the** 100 **task instances** from the development split of SWE-bench M (91 task instances have 1 issue, 9 have 2).

| Category | Count | Examples |
|---|---|---|
| Bug | 52 | "type: bug" (21); "[Type] Bug" (17); "bug" (14); |
| Feature | 45 | "[Type] Enhancement" (9); "[Feature] Signup & Account Creation" (6); "[Feature] Plans & Upgrades" (6); |
| Other | 89 | "Area:WebGL" (13); "L1 - broken" (7); "released" (6); "L2 - annoying" (5); "Store" (5); "FixTheFlows" (5); |

Mirroring Table 10, we redo the analysis of the distribution of tags for issues associated with task instances from the development set in Table 11. The findings for the development set are similar to the conclusions drawn from the test set. Bug fixes constitute a quarter of all found tags. The majority of tags, categorized under "Other", are specifically to subsets of 1+ repositories and reflect more fine-grained, interesting purposes. For example, the "Area:WebGL" tag, which is specific to `p5.js`, refers to problems that arise when using `WebGL`, a popular API for rendering 2/3-D graphics, to render `p5.js` components. Such problems require visual reasoning to fully comprehend; usually, the problem is related to how the component is created properly when using the default `P2D` renderer, but not when `WebGL` is used instead.

A.4    LICENSING

We fully list the license associated with each repository included in the SWE-bench M dataset in Table 12. These licenses all allow for non-commercial use of a repository's code and content, which the collection and evaluation processes of SWE-bench M respects.

Table 12: **Summary and licenses for all GitHub repositories represented in SWE-bench M**. Test split repositories are listed above the delimiter, while development split repositories are listed below.

| Repository | Summary | License |
|---|---|---|
| carbon-design-system/carbon | A design system built by IBM | Apache-2.0 |
| GoogleChrome/lighthouse | Automated auditing, perf. metrics for web | Apache-2.0 |
| grommet/grommet | React-based framework for web app dev. | Apache-2.0 |
| openlayers/openlayers | High-performance, feature-packed library for all your mapping needs | BSD-2-Clause |
| highlightjs/highlight.js | JS syntax highlighter with lang. auto-detection | BSD-3-Clause |
| scratchfoundation/scratch-gui | GUI to create + run Scratch 3.0 projects | BSD-3-Clause |
| bpmn-io/bpmn-js | BPMN 2.0 rendering toolkit + web modeler | Custom |
| quarto-dev/quarto-cli | Open-source scientific and technical publishing system built on Pandoc. | Custom |
| alibaba-fusion/next | Configurable component library for web | MIT |
| eslint/eslint | Find and fix problems in JS code. | MIT |
| PrismJS/prism | Lightweight, robust syntax highlighter | MIT |
| prettier/prettier | Prettier is an opinionated code formatter. | MIT |
| markedjs/marked | Markdown parser and compiler. | Custom |
| Automattic/wp-calypso | JavaScript and API powered WordPress.com | GPL-2.0 |
| processing/p5.js | JS library for learning to code, create art | LGPL-2.1 |
| chartjs/Chart.js | Simple HTML5 Charts w/ <canvas> tag | MIT |
| diegomura/react-pdf | Create PDF files using React | MIT |

# B COLLECTION

Additional information about new dataset curation processes introduced for SWE-bench M are included here, such as the reasons for and number of task instances removed due to inconsistency testing, challenges of setting up automatic evaluations for JavaScript repositories, and the procedure with which digital assets were automatically downloaded.

## B.1 STATISTICS

In Table 13, we provide a summary of the number of pull requests that were eventually successfully converted into viable SWE-bench M task instances along with how many pull requests were removed per step of the filtering process.

Table 13: **Per repository, the number of pull requests eventually converted into viable SWE-bench M task instances and task instances kept per stage of the filtering process.** The red subscript denotes the number of task instances that were filtered out at this stage with respect to the total from the previous stage.

| | PRs Crawled | Conversion | Validation | Inconsistent Test | Manual Filter |
|---|---|---|---|---|---|
| GoogleChrome/lighthouse | 6,022 | $116_{\downarrow 5906}$ | $55_{\downarrow 61}$ | $55_{-0}$ | $54_{\downarrow 1}$ |
| PrismJS/prism | 1,942 | $52_{\downarrow 1890}$ | $42_{\downarrow 10}$ | $38_{\downarrow 4}$ | $38_{-0}$ |
| alibaba-fusion/next | 2,133 | $85_{\downarrow 2048}$ | $50_{\downarrow 35}$ | $50_{-0}$ | $39_{\downarrow 11}$ |
| bpmn-io/bpmn-js | 820 | $86_{\downarrow 734}$ | $54_{\downarrow 32}$ | $54_{-0}$ | $54_{-0}$ |
| carbon-design-system/carbon | 8,096 | $196_{\downarrow 7900}$ | $143_{\downarrow 53}$ | $134_{\downarrow 9}$ | $134_{-0}$ |
| eslint/eslint | 7,454 | $42_{\downarrow 7412}$ | $11_{\downarrow 31}$ | $11_{-0}$ | $11_{-0}$ |
| grommet/grommet | 3,848 | $68_{\downarrow 3780}$ | $22_{\downarrow 46}$ | $22_{-0}$ | $21_{\downarrow 1}$ |
| highlightjs/highlight.js | 1,955 | $60_{\downarrow 1895}$ | $39_{\downarrow 21}$ | $39_{-0}$ | $39_{-0}$ |
| openlayers/openlayers | 9,466 | $143_{\downarrow 9323}$ | $88_{\downarrow 55}$ | $80_{\downarrow 8}$ | $79_{\downarrow 1}$ |
| prettier/prettier | 9,424 | $34_{\downarrow 9390}$ | $13_{\downarrow 21}$ | $13_{-0}$ | $13_{-0}$ |
| quarto-dev/quarto-cli | 1,620 | $60_{\downarrow 1560}$ | $25_{\downarrow 35}$ | $25_{-0}$ | $24_{\downarrow 1}$ |
| scratchfoundation/scratch-gui | 6,347 | $38_{\downarrow 6309}$ | $11_{\downarrow 27}$ | $11_{-0}$ | $11_{-0}$ |
| Automattic/wp-calypso | 61,165 | $214_{\downarrow 60951}$ | $37_{\downarrow 177}$ | $37_{-0}$ | $37_{-0}$ |
| chartjs/Chart.js | 3,792 | $200_{\downarrow 3592}$ | $33_{\downarrow 167}$ | $24_{\downarrow 9}$ | $24_{-0}$ |
| diegomura/react-pdf | 784 | $22_{\downarrow 762}$ | $11_{\downarrow 11}$ | $11_{-0}$ | $11_{-0}$ |
| markedjs/marked | 1,816 | $16_{\downarrow 1800}$ | $14_{\downarrow 2}$ | $14_{-0}$ | $14_{-0}$ |
| processing/p5.js | 3,186 | $42_{\downarrow 3144}$ | $27_{\downarrow 15}$ | $16_{\downarrow 11}$ | $16_{-0}$ |
| **Total** | 134,866 | $1{,}478_{\downarrow 133388}$ | $679_{\downarrow 799}$ | $643_{\downarrow 36}$ | $619_{\downarrow 24}$ |

## B.2 INCONSISTENCY TESTING

All task instances in SWE-bench M should yield consistent test case results, so we use a simple scheme to filter out candidate task instances that are inconsistent. For each task instance, we run the test cases five consecutive times and check whether 1) all runs contain the same test cases, and 2) all runs contain the same passing test cases and failing test cases. If either condition is false, we filter out the task instance.

## B.3 RESOURCE COLLECTION

Besides text, issue descriptions can contain

1. Images, often in the form of web browser screenshots detailing bugs or specifications of new features. For issues describing unexpected user interface behaviors that involve complex mouse actions (dragging, etc.) or a longer sequence of reproduction steps, animated images (GIFs) are often used.

2. Links to online integrated development environments (IDEs) for testing and showcasing HTML, CSS, and JavaScript code snippets. Around 17% of the instances contain at least one such link, and 15% contain more than one.

While images can be directly downloaded, copying the resources from the online IDEs is more involved. We handle four different such IDEs, *codesandbox.io* (63), *jsfiddle.net* (25), *codepen.io* (21), *stackblitz.com* (5), and *editor.p5js.org* where numbers indicate the number of occurrences in the final dataset.

The IDEs typically consist of three panes for the code in the three respective languages and an additional preview pane. Furthermore, menus provide options to link external resources, such as JavaScript and CSS libraries, from content delivery networks. Depending on the occurrence of the IDE, we have written extraction scripts (downloading and parsing source code or using web browser automation tools) or manually downloaded content to obtain all resources. Most snippets are a combination of HTML, CSS, and JavaScript, while others take the form of React apps. For the former category, the online IDEs apply a variety of heuristics to interpret the HTML (linking external resources, adding missing tags, etc.), so we similarly apply a series of postprocessing steps to ensure that the websites can be immediately served without additional steps.

## C  EXPERIMENTS

In this section, we include additional details concerning the experiments and ablations we ran on the SWE-bench M dataset. We provide information such as how the optimal context window for retrieval augmented generation was selected and the refactoring efforts required for making software development agent scaffolds compatible with SWE-bench M task instances. We also show more detailed analyses of experimental results, such as solve rates by repository and year.

### C.1  BASELINE ADAPTATIONS

Here we provide further technical details on how we adapted each of the software development scaffolds for evaluation on SWE-bench MM.

**Retrieval Augmented Generation.** We largely inherit the retrieval pipeline, prompts, and formatting for RAG from those used in Jimenez et al. (2024a). This includes building a standard BM25 index using pyserini. Adaptations include the inclusion of "reproduction code" where present referenced in links as described in Appendix B.3 and adapting the patch generation example in the prompt from a Python example to an equivalent JavaScript example.

Lastly, we perform some minor reformatting of issue text. We convert all images to Markdown-style links with the following format:

```
![alt text – or image url if not provided](image url)
```

**Agentless** (Xia et al., 2024) is split into two main phases, localization and repair, the former of which is heavily dependent on Python-specific constructs. The original localization phase relies on the Python `ast` module to collect the line numbers of all function and class declarations. Unfortunately, `ast` is specific to the Python abstract syntax grammar, and therefore cannot be applied to other languages such as Typescript and Javascript. We also found that the `Esprima` library for parsing Javascript code had issues with Typescript code, so we used the Tree-sitter parsing library, which supports a wide variety of languages. Due to differences in function variants in Python vs. Javascript (e.g. arrow functions in Javascript), we also added extra parsing steps in the localization step. Furthermore, Agentless uses several in-context Python code snippets for both the localization and repair phases – we replace all prompts with Javascript equivalents. Finally, when we provide the repository structure as input to the model, we replace their tab-indented nested repository structure with full file paths – we found this to be extremely important in ensuring the models do not hallucinate false file paths.

The original Agentless work generates a list of candidate solutions and manually checks whether they 1) pass syntax checks and 2) pass all *pass-to-pass* P2P tests. We do not include these filters for our implementation of Agentless because 1) we found syntax parsing to be inconsistent between

different versions of Javascript, leading to many valid candidate solutions being filtered unless a particular parser was used per instance and 2) we were interested in a fully automated framework. Finally, the original Agentless implementation did not implement support for Claude models, including how to handle extraneous outputs when querying for structured outputs. We added extra functionality to support Claude outputs for our experiments while remaining faithful to the original implementation.

**Aider** (Gauthier, 2024) similarly parses an abstract syntax tree for Python code to help the language model localize code with a limited context window – however, they use Tree-sitter instead of the `ast` module, enabling their framework to generalize easier to other languages. Other than adding the Javascript / Typescript versions of Tree-sitter, we do not modify Aider. Unlike Agentless, which simultaneously generates multiple candidates and does not require a syntax-checking step, Aider uses syntax-checking and linting in an iterative fashion to produce plausible generations. We found that due to syntax differences in Javascript versions, the parser would often invalidate correct or syntactically valid solutions generated by the language model, leading us not to use the baseline in its current form.

**AutoCodeRover** (Zhang et al., 2024b) also uses the `ast` module to generate an abstract syntax tree (AST) for Python code in its *context-retrieval phase*. Furthermore, it defines a set of APIs that the agent can use to traverse the AST. In our attempted implementation, we replace the `ast` module with the Tree-sitter library and some extra custom functions, and re-define the APIs to interface with the Javascript AST while retaining the same stratified search functionality as the original method. We found the traversal APIs to be too specific to the `ast` module, as the structure of the Tree-sitter Javascript AST was entirely different, leading to faulty localization.

To re-purpose AutoCodeRover for SWE-Bench M, a faithful implementation would require entirely swapping out the localization / code-searching APIs.

**Moatless** (Örwall, 2024) employs a multi stage pipeline for solving task instances. Similar to Agentless, Moatless also generates an AST of the desired language, and converts this AST into a custom code graph object that they use for their search and localization APIs. Because this code graph object is, in theory, language agnostic, it enables flexibility in applying Moatless to different programming languages. However, while the authors wrote code for converting a Python-generated AST to their code graph, we noticed that performing an analogous conversion for Javascript/Typescript does not capture all of the relevant design patterns (e.g. arrow functions). Thus, we chose not to use Moatless as a baseline.

**SWE-agent Base.** SWE-agent involves relatively few Python specific tools and components. The most notable exception to this is that for tasks in the original SWE-bench dataset, it sets up the OS environment with the necessary dependencies and installs the packages locally to provide the agent with a local testbed for experimenting with changes. To enable similar behavior in Javascript, we also install package dependencies and build the project locally when applicable. We also globally link the package name to the local repository when the instance contains an npm package.

New for SWE-bench M, we supply agents with "reproduction code" from links available in the problem statement. As described in Appendix B.3, these links and assets can contain useful code for reproducing issues locally. For each link containing reproduction code, we create a directory in the repository directory where the agent starts, and paste the code into the corresponding files (usually 'index.html', 'script.js', and 'style.css'). Finally, we initialize those repositories as an npm package and link the package from main repository directory so the agent may use the package as installed locally.

**SWE-agent JS.** Extending SWE-agent Base, SWE-agent JS adapts the base edit command to work more effectively with JavaScript and related programming languages. In particular, the original SWE-agent edit command includes a component that performs an linting operation after every edit an agent applies to a Python file. This linting operation checks whether the resulting file would be syntactically correct and can detect certain other mistakes such as a reference to an undnefined variable. During editing, SWE-agent will "roll-back" an edit if it is found to be invalid according to the linting operation. It then reports this error to the agent and requests that the agent to correct and try its intended edit again.

Table 14: Specialized tools provided to the agent for web interaction and image handling. Required arguments are enclosed in `<>` and optional arguments are in `[]`.

| Command | Documentation |
|---|---|
| **open_webpage** `<website_dir>` | Opens the webpage in a new display window using Google Chrome. `website_dir` is the directory where an index.html file exists and will be served. |
| **restart_webpage** `<website_dir>` | Restarts the webpage with the given directory. `website_dir` is the directory where an index.html file exists and will be served. |
| **close_webpage** | Closes the open webpage. |
| **screenshot** `[<num_images>]` `[<interval>]` | Takes screenshots of the display. `num_images` (optional, default 1) is the number of screenshots to take. `interval` (optional, default 1.0) is the interval between screenshots in seconds. |
| **open_image** `<image_path>` `[<image_path> ...]` | Opens the image(s) in base64 format to be fed to the LM. Multiple image paths can be provided. |

Figure 6: Action space of SWE-agent with different configurations for Claude 3.5 Sonnet and GPT-4o using the three configurations mentioned in Section 4 on the development set. For a version of this figure stratified by repositories, see Figure 7.

To adapt this for SWE-bench M, we use ESLint with the local repositories configuration files, to primarily detect fundamental syntax errors. While typically ESLint is capable of enforcing a wide variety of rules, we found that relying too heavily on these rules from the projects' configuration substantially degraded performance.

**SWE-agent M.** Further extending SWE-agent JS, SWE-agent M introduces a variety of new commands and features ot the ACI, which we show in Table 14. These new commands require some environmental support for simulating displays. We simulate a display for the X window system using Xvfb, and take screenshots with xwd.

## C.2 BASELINE CONFIGURATIONS

**SWE-agent.** Due to the expense and flexibilty in developing new SWE-agent interfaces, we developed new features and for SWE-agent JS and SWE-agent M primarily iterating while validating performance on a small subset of the development set. By default, for long trajectories, SWE-agent "collapses" past environment observations by replacing the content of old observations with a single line: "n lines omitted". SWE-agent will usually show the model the last five observations, and collapse all observations prior to that except the initial problem statement and demonstration. For the final configuration evaluated for each system, we perform a very small grid search over two different options for the number of past observations to show in $\{5, 9\}$. We perform this search on a set of 50 instances from the development set and show the results of this search in Table 15.

**Retrieval Augmented Generation.** For RAG systems, the amount of context we provide, either in terms of the number of documents or simply the absolute length of the context retrieved, is an important hyperparameter that may affect a model's performance. As in Jimenez et al. (2024a), we determine the final RAG system to evaluate by performing a grid search over three possible context lengths in $\{32K, 64K, 100K\}$ and the inclusion or not of images as input with the problem

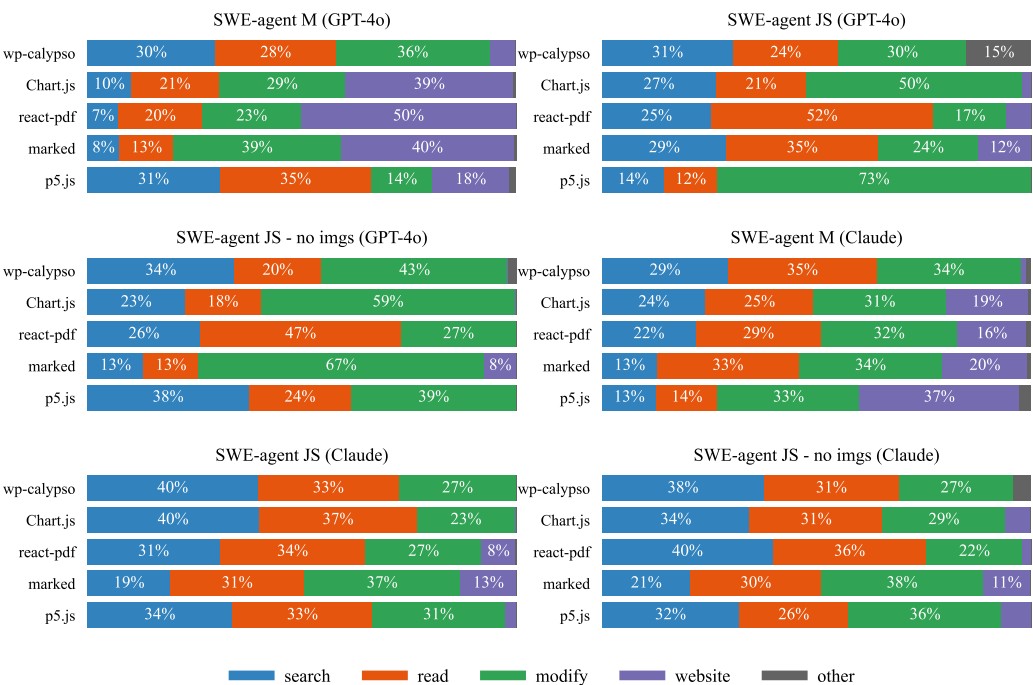

Figure 7: Action space of SWE-agent with different configurations for Claude 3.5 Sonnet and GPT-4o using the three configurations mentioned in Section 4 on the development set broken down by repository.

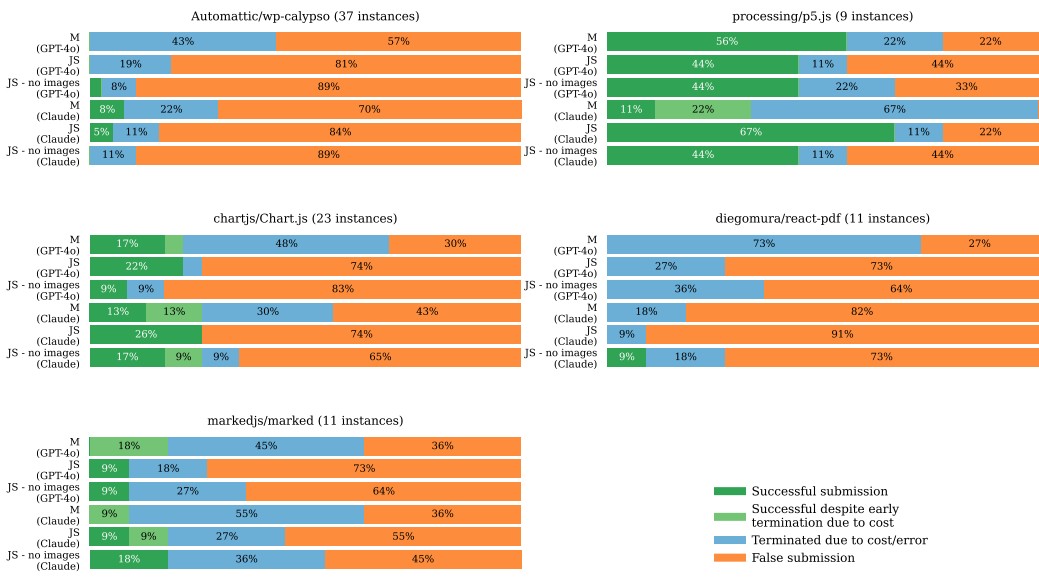

Figure 8: Frequency of outcomes by success and exit status of SWE-agent under different configurations on the development set.

Table 15: Hyperparameter search for SWE-agent system for non-collapsed history length.

| Model | System | History Length | % Resolved |
|-------|--------|:-:|:-:|
| Claude 3.5 Sonnet | SWE-agent Base | 5
9 | **18**%
14% |
| | SWE-agent JS | 5
9 | **28**%
18% |
| | SWE-agent M | 5
9 | **24**%
16% |
| GPT-4o | SWE-agent Base | 5
9 | **18**%
12% |
| | SWE-agent JS | 5
9 | **24**%
10% |
| | SWE-agent M | 5
9 | **26**%
16% |

Table 16: Hyperparameter search for RAG models with differing lengths of context and image input.

| Model | Retrieval Context | With Images | % Resolved |
|-------|:-:|:-:|:-:|
| Claude 3.5 Sonnet | 32K | ✗
✓ | $9.4_{\pm 1.1}$
$7.8_{\pm 2.6}$ |
| | 64K | ✗
✓ | $11.2_{\pm 1.3}$
$\mathbf{14.1}_{\pm 2.6}$ |
| | 100K | ✗
✓ | $7.6_{\pm 2.7}$
$13.7_{\pm 1.8}$ |
| GPT-4o | 32K | ✗
✓ | $6.7_{\pm 2.1}$
$6.9_{\pm 2.0}$ |
| | 64K | ✗
✓ | $6.7_{\pm 1.9}$
$7.6_{\pm 2.2}$ |
| | 100K | ✗
✓ | $8.0_{\pm 1.5}$
$\mathbf{10.0}_{\pm 2.2}$ |

statement. We run each configuration 5 times on the development set and select the configuration with the best mean performance. We report the results of this grid search in Table 16, highlighting the performance of our selected configuration.

## C.3 FURTHER ANALYSES

Here, we include additional analyses of the experimental results and performance by different baselines on SWE-bench M.

**Resolved by Repository.** We show the resolved rates per repository accomplished by different baselines in Table 17.

Table 17: **% Resolved Rates by repository for different baselines.** All performance numbers listed here use GPT-4o as the base model. Performance for the test split repositories (above delimiter) are pass@1, while performance for the development split repositories (below delimiter) represents the best result achieved from a hyperparameter search.

| Repository | Count | SWE-agent M | Agentless JS | RAG |
|---|---|---|---|---|
| GoogleChrome/lighthouse | 54 | 5.6 | 7.4 | 1.9 |
| PrismJS/prism | 38 | 0.0 | 0.0 | 0.0 |
| alibaba-fusion/next | 39 | 0.0 | 2.6 | 0.0 |
| bpmn-io/bpmn-js | 54 | 27.8 | 11.1 | 13.0 |
| carbon-design-system/carbon | 134 | 1.5 | 0.7 | 0.0 |
| eslint/eslint | 11 | 0.0 | 0.0 | 0.0 |
| grommet/grommet | 21 | 0.0 | 0.0 | 0.0 |
| highlightjs/highlight.js | 39 | 2.6 | 2.6 | 0.0 |
| openlayers/openlayers | 79 | 51.9 | 3.8 | 30.4 |
| prettier/prettier | 13 | 7.7 | 0.0 | 0.0 |
| quarto-dev/quarto-cli | 24 | 0.0 | 0.0 | 0.0 |
| scratchfoundation/scratch-gui | 11 | 0.0 | 0.0 | 0.0 |
| Automattic/wp-calypso | 37 | 0.0 | 0.0 | 2.2 |
| chartjs/Chart.js | 24 | 20.8 | 0.0 | 11.7 |
| diegomura/react-pdf | 11 | 0.0 | 0.0 | 0.0 |
| markedjs/marked | 14 | 7.1 | 0.0 | 20.0 |
| processing/p5.js | 16 | 25.0 | 6.2 | 23.8 |

**Temporal analysis does not reveal any indication of solution leakage.** Table 18 shows resolution rates for different baselines by year. While agentless with GPT-4o shows somewhat increased performance for older task instances, all other system shows significantly higher performance on newer instances with a pronounced peak in 2024. However, the instances of the different repositories have uneven temporal distributions and average resolution rates (see Tables 7 and 17), making it difficult to interpret the yearly solution rates. For example, 8 out of the 13 instances from 2024 are from `openlayers/openlayers`, which has by far the highest solution rates with SWE-agent M and agentless across all years.

To account for this, we calculate average resolution rates before and after the GPT-4o training cutoff (October 2023). We then compare the post resolution rates to the pre-resolution rates reweighted to the post-cutoff repository distribution[4]. The results are shown in Table 19 and show that the post-cutoff performance of all tested systems exceeds that of the pre-cutoff even when accounting for the shift in repository distribution.

Since Claude 3.5 Sonnet's training cutoff of April 2024 would leave too few instances, a similar analysis is not feasible. However, SWE-agent M with Claude 3.5 Sonnet follows a very similar trend as SWE-agent M with GPT-4o and shows an even more pronounced performance jump in 2024.

Table 18: **% Resolved performance for SWE-bench M test split task instances from different years.** Each row corresponds to a year, while each column is the agent's performance for task instances from that year.

| Year | Count | SWE-agent M (Claude) | SWE-agent M (GPT-4o) | Agentless JS (GPT-4o) | RAG (GPT-4o) |
|------|-------|----------------------|----------------------|-----------------------|--------------|
| 2017 | 22    | 4.5                  | 4.5                  | 9.1                   | 4.5          |
| 2018 | 31    | 0.0                  | 0.0                  | 3.2                   | 0.0          |
| 2019 | 96    | 9.4                  | 7.3                  | 3.1                   | 1.0          |
| 2020 | 91    | 8.8                  | 4.4                  | 5.5                   | 2.2          |
| 2021 | 103   | 13.6                 | 11.7                 | 2.9                   | 3.9          |
| 2022 | 101   | 12.9                 | 19.8                 | 1.0                   | 13.9         |
| 2023 | 60    | 13.3                 | 20.0                 | 0.0                   | 6.7          |
| 2024 | 13    | 46.2                 | 53.8                 | 7.7                   | 46.2         |

Table 19: **% Resolved Rates with GPT-4o before and after training cutoff.** This table shows the performance for the SWE-bench M test split task instances before and after the GPT-4o training cutoff date (October 2023). Pre-cutoff (reweighted) is the pre-cutoff performance reweighted to the post-cutoff repository distribution (see text).

|                          | Count | SWE-agent M | Agentless JS | RAG  |
|--------------------------|-------|-------------|--------------|------|
| Pre-cutoff               | 500   | 11.0        | 3.0          | 5.0  |
| Pre-cutoff (reweighted)  | 500   | 27.6        | 3.1          | 13.6 |
| Post-cutoff              | 17    | 47.1        | 5.9          | 41.2 |

---

[4]i.e., we calculate $\sum_{\text{repo}} f_{\text{post}}^{\text{repo}} \eta_{\text{pre}}^{\text{repo}}$, where $f_{\text{post}}^{\text{repo}}$ is the fraction of instances of a specific repository among the post-cutoff instances and $\eta_{\text{pre}}^{\text{repo}}$ is the corresponding pre-cutoff solution rate.

# D    HUMAN VALIDATION

We provide full details about the human validation process for SWE-bench M task instances. We include each question with corresponding examples that were provided to the papers' authors. We also show additional analyses about the authors' answers and agreement.

---

## D.1    IMAGE CATEGORIZATION

This question aims to categorize the types of visual content commonly associated with open-source software issues. By classifying images across a diverse range of repositories, we aim to analyze the visual ways in which developers convey information about software development. To accomplish this, we randomly sampled 50 images from the dataset and manually derived 8 categories from these samples. Human participants then manually inspected and labeled all images with one of the categories.

...................................................................................................................

*Prompt*: Classify the image as one of these categories: (1) Code Snippet Screenshot (2) Web Interface (UI/UX Element) (3) Map/Geospatial Visualization (4) Diagram (5) Data Visualization (Plots) (6) Artwork / Photography (7) Error Message (8) Miscellaneous.

Here are examples of each category of images.

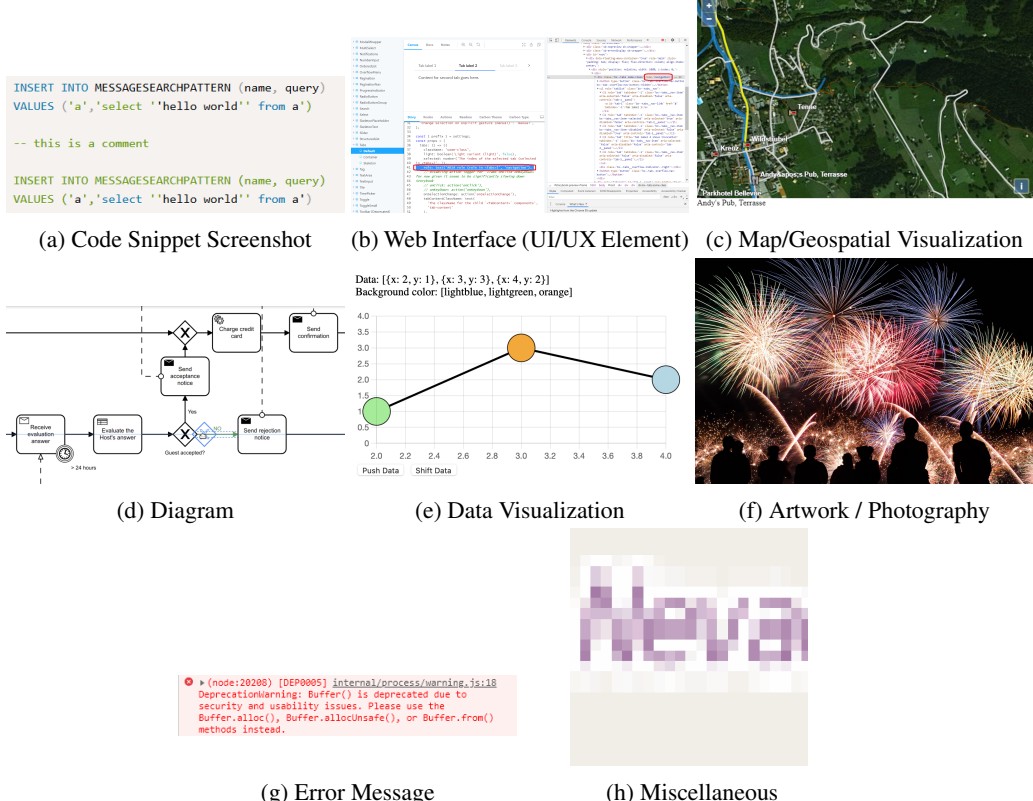

   (a) Code Snippet Screenshot     (b) Web Interface (UI/UX Element)     (c) Map/Geospatial Visualization

      (d) Diagram           (e) Data Visualization       (f) Artwork / Photography

          (g) Error Message            (h) Miscellaneous

The images are from the following task instances:

1. Code: `PrismJS_prism-1500`
2. Webpage: `carbon-design-system_carbon-6964`
3. Geospatial: `openlayers_openlayers-10545`
4. Diagram: `bpmn-io_bpmn-js-1542`
5. Data Viz.: `chartjs_Chart.js-9101`
6. Art: `quarto-dev_quarto-cli-5547`
7. Error Trace: `diegomura_react-pdf-1285`
8. Other: `openlayers_openlayers-12393`

Table 20: Response to *Image Categorization* indicates the majority of images are code and website screenshots. Other categories tend to be more repository-specific.

| | Code | Website | Geospatial | Diagram | Data Viz. | Art | Errors | Other | Total |
|---|---|---|---|---|---|---|---|---|---|
| bpmn-js | 1 | 13 | 1 | 54 | 1 | 1 | 1 | 1 | 69 |
| carbon | 21 | 136 | 0 | 36 | 0 | 0 | 30 | 1 | 224 |
| Chart.js | 5 | 2 | 0 | 0 | 18 | 0 | 1 | 0 | 26 |
| eslint | 10 | 0 | 0 | 1 | 0 | 0 | 2 | 0 | 13 |
| grommet | 3 | 25 | 0 | 2 | 2 | 0 | 0 | 0 | 32 |
| highlight.js | 62 | 4 | 0 | 4 | 0 | 0 | 0 | 0 | 70 |
| lighthouse | 5 | 55 | 0 | 0 | 5 | 0 | 3 | 0 | 68 |
| marked | 1 | 22 | 0 | 0 | 0 | 2 | 0 | 0 | 25 |
| next | 11 | 36 | 0 | 0 | 0 | 0 | 6 | 0 | 53 |
| openlayers | 6 | 6 | 35 | 0 | 0 | 0 | 2 | 0 | 49 |
| p5.js | 5 | 0 | 0 | 3 | 0 | 22 | 2 | 0 | 32 |
| prettier | 9 | 3 | 0 | 0 | 1 | 3 | 0 | 0 | 16 |
| prism | 41 | 3 | 0 | 3 | 0 | 0 | 0 | 0 | 47 |
| quarto-cli | 7 | 22 | 0 | 2 | 1 | 10 | 0 | 1 | 43 |
| react-pdf | 6 | 7 | 0 | 2 | 1 | 0 | 4 | 0 | 20 |
| scratch-gui | 0 | 12 | 0 | 0 | 0 | 1 | 0 | 0 | 13 |
| wp-calypso | 1 | 55 | 0 | 0 | 0 | 0 | 3 | 0 | 59 |
| Total | 194 | 401 | 35 | 107 | 28 | 38 | 54 | 2 | 859 |

..................................................................................................................................

*Results*: We applied this labeling procedure to the 862 problem statement images across all SWE-bench M task instances. The counts for each of the categories are presented below in Table 20.

The majority of SWE-bench M images are screenshots of webpages or code. These categories dominate because they are most often used to point out problems with linting (e.g. `eslint`, `highlight.js`, `prism`, prominently feature code-related images due to their focus on code analysis and syntax highlighting) and incorrectly rendered web elements (e.g. `carbon`, `lighthouse`, `next`, `wp-calypso`, as these tools are used to address issues in web design, layout and performance). The representation for other categories correlates heavily with specific repositories. In other words, images falling under a certain category are usually distributed across at most two to three repositories. For example, diagrams tend to be found in `bpmn-js`, a tool for creating BPMN (Business Process Model and Notation) diagrams, and `carbon`, which often uses design component specifications to communicate details about web elements, such as spacing and size.

Geospatial images, including maps or location-based data, are exclusive found in `openlayers`. Data visualization, such as charts or graphs, are predominantly found in the `Chart.js` repository, which specializes in creating data-driven visual content. Additionally, creative coding repositories like `p5.js` often include images related to artistic outputs, those images typically include visual glitches or unexpected rendering behaviors. Lastly, categories such as errors are found sporadically across repositories and represent instances where tools fail to operate correctly, such as compilation failures. Based on these findings, we conclude that SWE-bench M features a wide range of images that capture various aspects of software development, with code and webpage screenshots dominating the dataset. At the same time, certain types of images are tightly coupled with the specific functionalities of individual repositories, reflecting the unique focus and purpose of each project.

## D.2 IS AN IMAGE REPRESENTABLE AS TEXT?

How important is it for a SWE-bench M image to be an image? To answer this, we attempt to quantify the proportion of images that are solely text. For these images, it may not be necessary for the image's information to be communicated visually. For instance, it's possible that a large proportion of images are screenshots of code or error messages, some of which could be represented

perfectly as text. If this is the case, the fact that the information is presented as image becomes somewhat trivial.

To clarify, we are not asking whether the image can be processed with Optical Character Recognition (OCR) or verbalized, as such approaches would still likely result in a loss of information and are hard to answer without considering the problem context. For instance, for syntax highlighting libraries, although many images are code screenshots, there could be color highlighting of different characters that is crucial to understanding the problem correctly. Therefore, while traditional OCR could turn the code in the image into text, it would fail to capture additional visual details. Asking whether the same image can be verbalized is also not straightforward, as the answer could likely depend on the type of problem being solved and what information about the image is actually necessary. In a small labeling trial, we also found that answers are sensitive to the annotator's subjectivity, reflected by low inter-annotator agreement.

Similar to *Q1: Image Categorization*, we ask a human participant to look at the image and respond with either "Yes" if the image can be represented faithfully as text, and "No" otherwise. A more detailed discussion of how to decide is included in the following prompt.

.......................................................................................................................................................

*Prompt*: Can you assess if this image can be faithfully represented with text? A faithful representation means that the contents of the image can be represented as text with *no loss* in information. An image fits this criteria if (1) The image only contains text (2) The text in the image is mostly or entirely monochrome, and (3) The image does not contain any meaningful visuals or patterns independent of text that could be important to know for solving a problem. Please only respond with [1/0], 1 meaning "Yes" and 0 meaning "No". Do not include any additional justification or text.

Here are examples of images that can and cannot be conveyed effectively with OCR.

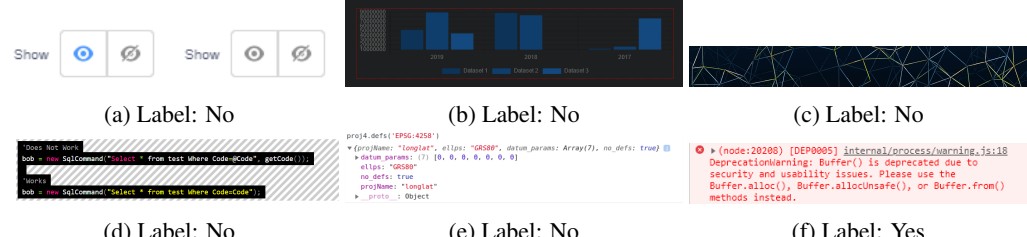

(a) Label: No           (b) Label: No           (c) Label: No

(d) Label: No           (e) Label: No           (f) Label: Yes

Figure 10: Examples of images that cannot (top row) and can (bottom row) be processed into text by OCR software. Typically, code and webpages are OCR-able while diagrams, plots, and art are not.

- Top Left (`scratchfoundation_scratch-gui-8891`): The eye icons cannot be represented as text. While there is text, it does not take up much of the image.
- Top Middle (`chartjs_Chart.js-8567`): The bar plot cannot be converted into text.
- Top Right (`processing_p5.js-3709`): This is an graphic with no text in it.
- Bottom Left (`openlayers_openlayers-11649`): Although a significant portion of this image is made up of text, the code syntax coloring is an aspect of the image that cannot be conveyed effectively in a text representation. In addition, the black highlighting and striped background cannot be portrayed as text.
- Bottom Middle (`diegomura_react-pdf-1285`): Similar to the bottom left image, this is also "No" because of the text coloring.
- Bottom Right (`PrismJS_prism-2782`): The contents of this image can be faithfully represented as text, as the majority of it is colored red. The main purpose of this screenshot is to communicate an error message. There are no meaningful visual elements.

.......................................................................................................................................................

*Results*: We again answer these questions for problem statement images in SWE-bench M. Out of 862 problem statement images, 691 (80%) were labeled as "No", while 171 (20%) were labeled as

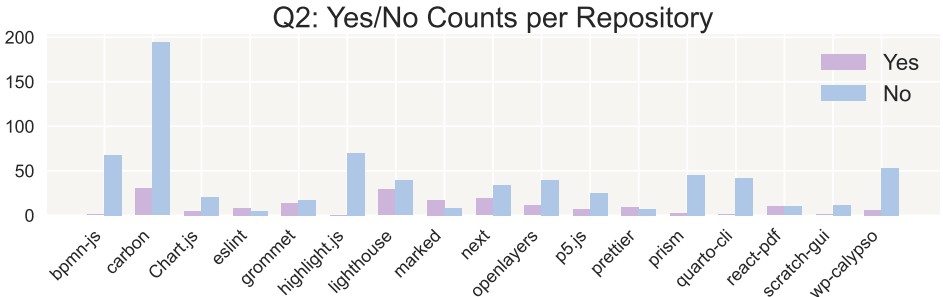

Figure 11: Responses to whether an image can be represented with text per repository.

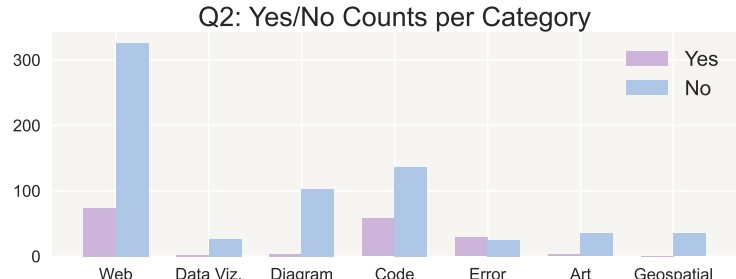

Figure 12: Responses to whether an image can be represented with text per image category.

"Yes". Below, we present two figures. Figure 11 shows the labeling results per repository. Figure 12 shows the labeling results with respect to image category.

With the exception of `prettier`, across repositories, the majority of images were labeled as not representable with text, with several repositories having no images labeled "Yes" (e.g. `bpmn-js`, `highlight.js`, `quarto-cli`). The principal component that corresponds to whether an image can be labeled is the image category. As shown in Table 12, different image categories have varying rates of whether or not the image can be represented with text. Error messages are more often labeled "Yes" than not. A smaller proportion of web and code screenshots were labeled "Yes". Although at first glance it may seem like this proportion should be higher, through the annotation procedure, we found out that for code, text color and highlighting are often very important signals for solving task instances, particularly linting repositories such as `prism` and `eslint`. For web screenshots, while there is usually text on a webpage, the spatial arrangement of web components is a critical aspect of understanding problems in design framework libraries, such as `carbon` and `next`. For the remaining categories, low to no images are labeled as being representable in text.

Based on our findings, we conclude that information from problem statement images in SWE-bench M frequently must be represented as images. The rate at which images are not just screenshots of text varies by repository and image category. Images that are errors are more often representable as text. Other categories are less often due to either a complete absence of text or the text coloring and organization being an important part of understanding the problem being communicated.

### D.3 IMAGE NECESSITY.

We elicit human feedback on how necessary images are to solving the task at hand. This question attempts to directly determine how the visual asset(s) provided with a problem statement are actually necessary to solving a GitHub issue when provided. We label the 557 SWE-bench M task instances that have one or more images in its problem statement.

For the annotation procedure, a human participant is asked to look at each task instance's problem statement, codebase, gold patch, and test patch, and judged whether it could be solved. They should

then look at the associated problem statement image(s) and assign a label according to the procedure described below. Task instances with no images in the problem statement are not considered for this question. If a task instance is deemed generally unsolvable, it is removed from the dataset.

...................................................................................................................................................

*Prompt*: For each task instance, please determine whether its associated images are necessary to solving the problem described by this task instance. The following workflow is recommended:

1. Read the problem statement. If you'd like, check the codebase, gold patch, and test patch as needed. Make a mental note of whether the task instance can be solved.

2. Then, look at the image(s) associated with the task instance. Now, re-answer whether the task instance can be solved.

3. Provide answer according to the following logic:
   - "Yes": If the answer changed from "No" to "Yes", then the image is necessary.
   - "No": If the answer remained "Yes", the image is not necessary to solve the task.
   - Remove: If the answer before and after seeing the image was both "No", please note this and we will remove the task instance.

Below, we provide examples of task instances where the image *is* necessary to solving the task (`PrismJS__prism-3442`), and where it *is not* (`quarto-dev__quarto-cli-6659`).

**PrismJS__prism-3442.** A screenshot of code is provided for this task instance. It is shown together with the corresponding problem statement in the text box below.

The main contribution of this image is the syntax highlighting. Specifically, it shows the undesired effect of Prism highlighting the quotes with the incorrect color. The double quotes around "Month", "Days", and "Jan" should not be highlighted gray.

Without the screenshot, it is not exactly clear how the quotes are being highlighted incorrectly. For instance, it's unclear what color corresponds to "highlighted as punctuation". In the absence of the image, while it is still possible that an experienced maintainer of this repository could make educated guesses about the exact nature of the bug or reproduce it exactly, the image provides concrete evidence of what the user is experiencing.

---

`PrismJS__prism-3442`: [language-markup] Quotes in HTML attribute values are highlighted

**Information:**
- Prism version: Latest (reproducible on test.html)
- Plugins: none
- Environment: Browser

Quotes are highlighted as punctuation inside HTML attributes. E.g. this:

```html
<google-chart data='[["Month", "Days"], ["Jan", 31]]'></google-chart>
```

is highlighted as:

```
<google-chart data='[["Month", "Days"], ["Jan", 31]]'></google-chart>
```

We might decide that this is a feature, not a bug, but figured I'd flag this in case, as it looked wrong to me.

---

On the other hand, the image in `quarto-dev__quarto-cli-6659`'s problem statement is not necessary.

The provided image is an arbitrary link that is used in the reproduction code shown in the issue description. While the image is helpful in that it is readily usable by a developer for recreating the problem, there is no additional information being conveyed by the image's content that helps with understanding the task at hand.

---

**`quarto-dev_quarto-cli-6659`: Figures from R code blocks don't render if 'fig-cap: !expr . . .' evaluates to 'character(0)'**

With Quarto 1.4.330, quarto-r 1.2, and knitr 1.43 on R 4.2.1, the command 'quarto render example.qmd –to html' renders the following Quarto document as expected:

```r
#| fig-cap: !expr caption
caption = "hello world"
knitr::include_graphics("http://arfer.net/mlp/img/rara-jiggs.png")
```
But if 'caption = "hello world"' is changed to 'caption = character(0)', the figure disappears from the output.

I originally hit this issue with a mistaken 'sprintf' call, which can produce a zero-length character vector.

---

*Results*: Out of 557 task instances, 465 were labeled as the image being *necessary* to solving the problem. The remaining 93 were labeled as "No", meaning the image is *not necessary*. The following Figure 13 shows the labeling splits for this question by repository.

Figure 13: Responses to whether an image is necessary to solve a task instance per repository.

Q3: Yes/No Counts per Repository

Annotations suggest the significant majority (83.5%) of problem statement images are necessary to solving the problem at hand. Several repositories have zero issues with unessential images.

Annotators attributed several factors that affected their decision making. First, if the task instance included reproduction code, annotators were more likely to rate the task instance as not requiring the image. For these cases, the image was redundant because running the reproduction code would produce the exact effect that the image was capturing. Second, a repository's contributing guidelines influences the role of the image. For instance, some repositories will explicitly ask for an image or screenshot that details the error message, which is why for repositories such as `bpmn-js` and `lighthouse`, there are zero occurrences of unessential images. On the other hand, images are sometimes provided under an "Additional Information" section, in which case the image is helpful, but not crucial. These observations also hold for reproduction code. Task instances from `carbon`, `p5.js`, and `next` will often have links to online editors reproducing the described bug.

### D.4 Task Difficulty

Following the instructions used in Chowdhury et al. (2024), we label the difficulty of a task instance. We define difficulty as the estimated amount of time it would take for an experienced developer to accomplish this task. There are four possible labels for difficulty:

- **<15 min** fix (e.g., a trivial change adding some assertions to a function)
- **15 min–1 hour** (e.g., a small change that requires a bit of thought)
- **1–4 hours** (e.g., substantially rewriting a function or editing multiple files)
- **>4 hours** (e.g., a very esoteric issue that clearly requires a substantial amount of research to fix, changing >100 lines of code)

For every estimate, we assume the developer is approaching the problem from the following context:

- The developer has neither worked on nor used the codebase before.
- The developer is familiar with JavaScript code and has experience working on JS related frameworks (e.g., React).
- The developer has had a couple hours to become familiar with the codebase prior to starting work on the task instance.

To label, each annotator looked at the full details of the task instance (problem statement, images, gold patch, codebase) and made a best-effort judgment of how intensive the fix would be. Annotators were asked to *not* consider the time spent on writing or modifying tests.

Because of the potential variance in annotators' answers to this question along with the amount of time it takes to comprehend a task instance before making a judgment, we do not label the entire SWE-bench M dataset. Instead, we take 100 task instances from all of SWE-bench M, uniformly sampling with respect to the ratios of task instances in each repository. Three annotators then label each of the tasks with one of the four labels. For each task instance, the final difficulty level is assigned to be whichever label has a majority vote. If there is no majority vote, the "middle" of the three annotations (either **15 min–1 hour** or **1-4 hours** is used as the label.

......................................................................................................................................

*Results*: From this labeling procedure, 100 SWE-bench M task instances are labeled with differing levels of difficulty, as shown in Table 21. The Fleiss' kappa score is calculated to be 0.78, reflecting a substantial amount of agreement between the three annotators who carried out labeling.

Table 21: Estimated level of difficulty for different splits of SWE-bench in addition to SWE-bench M. Each number is the percentage of sampled task instances classified at that level of difficulty.

| Dataset | # Samples | <15 min | 15 min-1 hour | 1-4 hours | >4 hours |
|---------|-----------|---------|---------------|-----------|----------|
| SWE-bench | 1,699 | 24.5 | 53.3 | 19.4 | 2.8 |
| Lite | 231 | 37.7 | 56.3 | 6.1 | 0.0 |
| Verified | 500 | 38.8 | 52.2 | 8.4 | 0.6 |
| SWE-bench M | 100 | 13.0 | 43.0 | 38.0 | 6.0 |

After the labeling procedure was completed, annotators agreed that the strongest signals for estimating the level of difficulty came most frequently from three indicators.

First, smaller gold patches tend be labeled as requiring less time and visa versa. For instance, for the 13 task instances labeled to take $< 15$ minutes, the gold patch change sizes (calculated as Lines Added + Lines Removed) are $[1, 3, 4, 1, 3, 2, 163, 5, 5, 10, 7, 6, 4]$. The one aberration of $\pm163$ lines is due to updates to an auto-generated `package-lock.json`. On the other hand, for task instances requiring $> 4$ hours to fix, the gold patch sizes are $[350, 186, 557, 367, 2682, 937]$.

Task instances with more descriptive problem statements tend to get rated as easier. The annotators' noted that more detailed descriptions and the presence of reproduction code made them more

inclined to reduce the estimated amount of time required. The images in the problem statement also had an effect on the labeling. Annotators mentioned that screenshots of web elements and source code helps developers localize errant code in a codebase faster. For instance in `carbon` and `next`, many screenshots would point out spacing or coloring issues with a specific web component. A screenshot is often informative for not only describing the problem, but also directly point out which component folder the task worker should look into.

Finally, annotators noticed that sometimes, a small change can still take some time, particularly if the edit is precise and requires a good amount of context to find. In addition to the size of a gold patch edit, the quality of an edit, specifically the types of entities and symbols that were used in the fix, influences the amount of time a problem takes. For instance, a larger edit that mostly consists of JavaScript primitives is usually regarded as simpler than a shorter edit mostly made up of entities from other parts of the codebase.

In conclusion, SWE-bench M presents task instances at a range of difficulties, from short quick fixes to problems that require large scale refactoring of multiple modules in a codebase. The difficulty distribution suggests that SWE-bench M can be effective for tracking improvements in model and agent capabilities.

## E    LIMITATIONS

**Broader scope**    To comprehensively evaluate multimodal AI systems, we curate 617 task instances from 17 JavaScript libraries for SWE-Bench M. However, the scope could be broadened across three key dimensions: **(1) More programming languages:** Besides JavaScript, multimodal content also appears in issues related to code in other programming languages, such as Python, C++, or Rust. **(2) More modalities:** While images and videos are likely the most commonly used media besides text in GitHub issues, users can also upload audio files or other media and file types. **(3) More tasks:** Our 617 tasks across 17 libraries cover various domains including web frameworks, data visualization and syntax highlighting, however, there are many more JavaScript libraries in other domains that could be added to the benchmark. We are excited about research broadening the scope across these and other axes. However, here we focus on quality over quantity. Scope extensions likely warrant separate dedicated future work, as we found that adding task instances to SWE-bench M was a labor-intensive task.

**Improved models and environments**    While we primarily focus on developing the benchmark and evaluating current models, we also extend SWE-Agent to handle multimodal issues. There remains substantial opportunity for future enhancements, both by advancing the underlying models (e.g., leveraging future versions of GPT and Claude) and by enriching the agent's environment with, for example, better browsing capabilities or more tools. We look forward to seeing future work along these lines to improve performance on SWE-Bench Multimodal beyond the systems that we present.

