# OpenReview forum: "SWE-bench Multimodal: Do AI Systems Generalize to Visual Software Domains?"
_ICLR.cc/2025/Conference — ICLR 2025 Poster_

### Official Review · Reviewer_v5Tx · 2024-10-30

**Soundness:** 2
**Presentation:** 3
**Contribution:** 2
**Rating:** 5
**Confidence:** 4

**Summary:**

The paper introduces the SWE-bench Multimodal (SWE-bench M), an extension of the SWE-bench benchmark for evaluating the performance of autonomous software engineering systems on their ability to fix bugs in visual, user-facing JavaScript projects. The dataset includes 619 task instances from 17 repositories, incorporating visual elements like screenshots, diagrams, and other assets to reflect real-world programming challenges more accurately. The authors also assess existing models developed for SWE-Bench on SWE-bench M, highlighting the limitations of current state-of-the-art approaches in handling visual information and cross-language generalization.

**Strengths:**

+ a new dataset for evaluating LLM-based issues resolving models
+ limitations of existing approaches on the new data were discussed

**Weaknesses:**

- criteria for project and issue selections are unclear
- generalizability of the JS-based multimodel issues is limited
- there already exists a rich research body on multimodal bugs/issues---mobile bugs/issues
- the adaptions for existing models are mainly on the language level, it's unclear how to handle images/videos
- the comparison between these approaches on multimodal issues and text issues is missing
- how important the visual data (images/videos) in the process of bug fixing is unknown

**Questions:**

1. This work extended SWE-Bench, while major challenges encountered during data collection may be similar to those in SWE-Bench. Could authors clarify what unique challenges were faced in data collection for this work compared to SWE-Bench?

2. There is an established research body on reproducing/fixing multimodal bugs in mobile apps, while this work focuses on JavaScript bugs. What is the motivation behind focusing specifically on JavaScript-based multimodal bugs, given there already exist mobile app bug datasets?

3. The data collection process inlcudes five steps, but the criteria for each step are unclear. For example, in step 1, the authors selected 17 repositories. What was the project sample size after applying the filters (e.g., 5,000 or more stars and 500 or more pull requests)? Step 5 includes human validation, how many authors involved? How to solve the conflict during the process?

4. "From 135k PRs, this new filtering criteria yields 1,478 candidate task instances." Does this indicate that the identified bugs are relatively uncommon or corner cases? How important is the importance of these collected multimodal bugs compared to text-only bugs?

5. Do the baseline models perform significantly differently on multimodal issues than text-only issues within these JavaScript projects?

6. The impact of visual data (images/videos) on the bug-fixing process is unclear; the performance decline without images/videos is around 3%. Does this imply that images/videos are unimportant for these tasks?

---

> ### Author Response · Authors · 2024-12-03
> **Response to Reviewer v5Tx (W1, Q1, W2, W3, Q2, W4)**
>
> We sincerely thank the reviewer for their feedback on our work.
>
> > W1: Criteria for project and issue selections are unclear
> > Q1: This work extended SWE-Bench, while major challenges encountered during data collection may be similar to those in SWE-Bench. Could authors clarify what unique challenges were faced in data collection for this work compared to SWE-Bench?
>
> Please refer to Section 2.2, where we address exactly this question and thoroughly review the project and issue selection criteria step by step. The primary challenges were namely adjusting task collection to work for JavaScript repositories (e.g. more heuristics that target image assets), designing entirely new execution environments from scratch, and performing extensive human validation (absent from SWE-bench).
>
> > W2: generalizability of the JS-based multimodal issues is limited.
>
> Our interpretation of this weakness is that JavaScript-based multimodal issues do not reflect all of Software Engineering. If this is correct, we agree, but the discussion around generalizability is less in reference to the coverage of this benchmark, and more in reference to how existing agent systems are designed. We make no mention of generalizability in Section 2, and only bring up generalizability in the context of AI system design in Section 3. In short, we are not claiming SWE-bench Multimodal represents all of software engineering, but the fact that there is such a significant performance gap between SWE-bench M and SWE-bench systems is very interesting because it calls into question generalizability.
>
> > W3: there already exists a rich research body on multimodal bugs/issues---mobile bugs/issues
> > Q2: There is an established research body on reproducing/fixing multimodal bugs in mobile apps, while this work focuses on JavaScript bugs. What is the motivation behind focusing specifically on JavaScript-based multimodal bugs, given there already exist mobile app bug datasets?
>
> We review prior works in the Related Works section. SWE-bench Multimodal is filling a clear research gap - the evaluation of AI systems on software engineering issues with visual components. There are several existing works on GUI navigation and multimodal HumanEval-style function completion, but to the best of our knowledge, there is no work that addresses editing a repository given visual specifications.
>
> If the reviewer can refer us to the mobile app bug datasets like SWE-bench Multimodal, we’d be happy to incorporate it into our Related Work section and Table 1. We are aware of works such as [1] [2] and [3], but these works are significantly different from SWE-bench Multimodal or repository-level code generation/editing, as they focus on automating user testing for mobile apps. These works evaluate how well a system can recognize or reproduce bugs in a mobile app. Code completion is a part of such works. It does not evaluate how an app’s codebase can be updated to resolve an issue.
>
> [1] Wendland, Tyler, et al. "Andror2: A dataset of manually-reproduced bug reports for android apps." 2021 IEEE/ACM 18th International Conference on Mining Software Repositories (MSR). IEEE, 2021.
>
> [2] Xiong, Yiheng, et al. "An empirical study of functional bugs in android apps." Proceedings of the 32nd ACM SIGSOFT International Symposium on Software Testing and Analysis. 2023.
>
> [3] Fazzini, Mattia, et al. "Enhancing mobile app bug reporting via real-time understanding of reproduction steps." IEEE Transactions on Software Engineering 49.3 (2022): 1246-1272.
>
> > W4: the adaptions for existing models are mainly on the language level, it's unclear how to handle images/videos
>
> We address these points directly in the main paper, although if the reviewer has more specific qualms, that would be helpful. Please refer to Section 3.1, where we discuss specifically how (1) existing systems were adjusted such that images + videos can be fed to the underlying LM, and (2) when possible, new tools were created to help with viewing, creating, and manipulating visual assets in a codebase or via a browser.

---

> ### Author Response · Authors · 2024-12-03
> **Response to Reviewer v5Tx (W5, W6, Q3-5)**
>
> > W5: the comparison between these approaches on multimodal issues and text issues is missing
>
> This is discussed in our report (Sections 2.2, 2.3, 3.1 + Table 5), although we would appreciate more clarification regarding what additional analyses the reviewer might be looking for.
>
> Sections 2.2 and 2.3 discuss the different challenges in collection strategy (simply, that SWE-bench Multimodal’s collection strategy is much more extensive than SWE-bench’s). Section 3.1 details how AI systems were adapted to incorporate visual feedback, and Table 5 + results illustrate how the inclusion of visual components in SWE-bench Multimodal makes for a more challenging benchmark. Our human annotation procedure (Appendix D) shows that SWE-bench M is generally more difficult based on estimated time to complete a task. Appendix A includes multiple breakdowns illustrating quantitative differences between SWE-bench Multimodal issues compared to SWE-bench issues (Figure 5).
>
> > W6: how important the visual data (images/videos) in the process of bug fixing is unknown
>
> We address this extensively. In Section 2.3, Appendix D.(1,2,3), we discuss how our manual validation process with human annotators strongly suggests that visual components provided in a naturally occurring GitHub issue are indispensable to solving the problem (83.5% of task instances were marked as unsolvable without its associated image(s)). These challenges are also echoed empirically in Section 4.1, where evaluation metrics and our analysis of trajectories continue to reflect these points.
>
> > Q3: The data collection process inlcudes five steps, but the criteria for each step are unclear. For example, in step 1, the authors selected 17 repositories. What was the project sample size after applying the filters (e.g., 5,000 or more stars and 500 or more pull requests)? Step 5 includes human validation, how many authors involved? How to solve the conflict during the process?
>
> - *What was the project sample size after applying filters?* - For specific metrics on how many candidate task instances were filtered out at each collection stage, please refer to Table 13. As you write, 135k PRs is filtered down to 1478 candidate task instances.
> - *How many authors involved* - 10 authors. We will update the paper to say this explicitly.
> - *How to solve conflicts during the process?* - In Appendix D, the explicit annotation + label selection processes are provided per question. In general, for easy questions (e.g. label image category), just one annotator was used per question. For harder questions (e.g. estimating task difficulty), four annotators provided labels for each question + task instance, and the final answer was selected via majority vote.
>
> > Q4: "From 135k PRs, this new filtering criteria yields 1,478 candidate task instances." Does this indicate that the identified bugs are relatively uncommon or corner cases? How important is the importance of these collected multimodal bugs compared to text-only bugs?
>
> The candidate task instances meet a number of criteria, including (1) the PR has 1+ issues (2) the PR has 1+ tests (3) either the test(s) or issue(s) have images in them. In SWE-bench, which put forth the first two criteria, usually just 10% of PRs passed this filter. In the case of SWE-bench Multimodal, criterion (3) filters out an additional 90% of SWE-bench style candidates, which is high yield (compared to nearly 99% being filtered out for Python, a number we determined in a minor experiment early on in the project).
>
> The implication that these filters may lead to a distribution shift that yields uncommon, non-representative task instances is a non-issue. Our extensive manual human validation of task instances, which involved the authors reading every single issue description, suggest that task instances are meaningful and visuals are necessary. From Figure 5, it’s also apparent that quantitatively, SWE-bench M instances have [problem statement length, files/functions/lines edited, # of tests distributions] that are similar to SWE-bench task instances.
>
> > Q5: Do the baseline models perform significantly differently on multimodal issues than text-only issues within these JavaScript projects?
>
> We agree this is an interesting question. We believe that we address this with our analysis that indicates how understanding visual components is one of two main reasons for why SWE-bench M is harder than SWE-bench (the other being JavaScript). To answer the reviewer’s version of this question, we would have to perform additional effort of curating a SWE-bench style dataset for JavaScript task instances, which is non-trivial. Curating SWE-bench “JS” is also tangential to the focus of our paper.

---

> ### Author Response · Authors · 2024-12-03
> **Response to Reviewer v5Tx (Q6)**
>
> > Q6: The impact of visual data (images/videos) on the bug-fixing process is unclear; the performance decline without images/videos is around 3%. Does this imply that images/videos are unimportant for these tasks?
>
> No, this does not imply images/videos are unimportant. Our analysis attributes the 3% improvement to images/videos. But the fact that the gain is only 3% suggests, rather, that existing systems do not utilize visual components effectively. Sections 2.2 and 2.3 emphasize, with human effort, that images/videos/visuals in an issue *are* necessary to solving the problem. Section 4.1 is our analysis point dedicated to showing that improvements on SWE-bench Multimodal performance correspond to improved abilities in image understanding.

---

### Official Review · Reviewer_yGjW · 2024-11-03

**Soundness:** 3
**Presentation:** 3
**Contribution:** 3
**Rating:** 6
**Confidence:** 3

**Summary:**

This work creates a new benchmark SWE-Bench Multimodal which address some limitations of prior benchmarks in this field that mainly focus on evaluation on Python only repositories. SWE-bench M provides a benchmark suite in JavaScript with the input  being to include also visual elements. SWE-bench M contains more than 600 different tasks complete with test cases to measure functional correctness. The authors further modify some existing state of the art open-source techniques and found that many of them require python-specific designs that limit their ability on the new benchmark

**Strengths:**

- Creates an important benchmark for an important area of measuring repository level code generation especially when the input is multi-modal
- Extensive manual effort used to ensure the benchmark is useable as well as adopt and modify baseline techniques

**Weaknesses:**

Overall, this is an important and usable benchmark, below are some weakness that I would love to see the authors address

**More analysis on the difficulty of the dataset**
- As the current demonstrate that SWE-bench M seems to be more diffcult than SWE-bench and SWE-bench lite, I would suggest the authors take a deeper look into the types of problems in SWE-bench and the reason why it is more difficult
- While the authors mentioned the mulit-modality aspect and the requirement of images, is that the only factor?
- It seems from looking at table 5 (the tasks without requiring images according to human annotators), the problems still appears to be more difficult that original SWE-bench lite problems
- More analysis into what makes these problem difficult would be beneficial to the research community

**Improvements to Testing**
- Given the state of software development is test driven, I would like to see the authors make some effort in improving the test quality.
- As mentioned in text, the authors already filter out some "flaky" tests which are a good sign, however looking at the number of tests in Figure 5 and Table 9 in Appendix it seems the number of Fail to Pass tests are still limited.
- It would be great if the authors can apply some test augmentation or manually apply (while this might require tons of manual effort, it woud definitely improve the robustness of the benchmark)

**More metrics than just pass or fail**
- One issue with repository level benchmarks (including SWE-bench) is that the metric used seems to be always whether the issue was resolved or not.
- For real-world software development, there are many steps to solving the problem from being able to identify the root cause, localize the code elements, writing tests that can reproduce the errors, and then finally applying the patch
- I believe that more intermediate metrics that can be used to gain more information from the results should be designed and applied.
- This would aid in the design of new approaches as well as provide more analysis options espeically given the large cost of evaluation on repository level benchmarks like SWE-bench M

**Questions:**

- Given the low amount of changes required to go from SWE-agent to SWE-agent JS and SWE-agent M, I am curious if the authors have tried applying SWE-agent JS or SWE-agent M on the original SWE-bench tasks? if so what would be the performance?
- How are the human annotators chosen? the paper only mentions the number of annotators (3) in the Appendix, but no further details are provided

---

> ### Author Response · Authors · 2024-12-03
> **Response to Reviewer yGjW (Weaknesses)**
>
> We sincerely thank the reviewer for their response (and for the friendly categorization of feedback).
>
> > W1: More analysis on the difficulty of the dataset
>
> The goal with Section 2.3 (Features) and Section 4.1 (Analyses) is to communicate (1) that SWE-bench Multimodal is more difficult for AI systems, and (2) this difficulty can be directly attributed to the necessity of visual understanding that existing VLMs struggle with. Our human annotation efforts are also meant to suggest how SWE-bench Multimodal problems are tractable for humans, but difficult for models because of the introduction of visual components.
>
> With respect to the reviewer’s feedback, we can understand how it might seem like we are suggesting that for humans, SWE-bench Multimodal problems are inherently harder than SWE-bench problems. This is not our goal, and we will update the paper such that it is clear we are *not* claiming this. In fact, we aim to show that SWE-bench Multimodal problems are very tractable for human developers. It would be prohibitively expensive for us to have human task workers attempt every instance, so following the work of SWE-bench Verified, we use human annotations (mainly Q.2 - Q.4) to demonstrate that (1) the tasks are do-able (anything considered impossible was removed), and (2) the difficulties and key to solving SWE-bench Multimodal lie in visual understanding.
>
> > W2: Improvements to Testing
>
> We agree with the reviewer’s point entirely - Test augmentation for SWE-bench Multimodal / SWE-bench would (1) greatly improve the quality of the benchmark, but (2) it is also difficult enough that it can constitute an entirely separate research contribution. (See SWT-Bench [1], which is similarly a derivative of SWE-bench that focuses exclusively on test generation).
>
> The main challenge of such an effort is that codebase unit tests are nowhere near as self-contained as traditional code generation (e.g. HumanEval) unit tests. In codebases, tests are often supported by an entire setup procedure of mocks, stubs, and testing entities that require understanding in order to write meaningful unit tests. These challenges are only complicated by the variance in testing infrastructures across repositories and time.
>
> As with the original SWE-bench work, we maintain faith in the scraped tests because they are written by a human maintainer, suggesting that the test is extremely faithful to the intended behavior. In its current form, SWE-bench Multimodal is already an effective evaluation for the same reasons as SWE-bench when it comes to testing (if not better due to the more extensive testing facilities that come with JavaScript + visual components as discussed in Section 2.3). Improving coverage like EvalPlus [2] did for HumanEval is great, but non trivial.
>
> [1] Mündler, Niels, et al. "Code Agents are State of the Art Software Testers." arXiv preprint arXiv:2406.12952 (2024).
>
> [2] Liu, Jiawei, et al. "Is your code generated by chatgpt really correct? rigorous evaluation of large language models for code generation." Advances in Neural Information Processing Systems 36 (2024).
>
> > W3: More metrics than just pass or fail
>
> We generally agree that introducing more metrics to repository-level coding tasks would be interesting. In our analyses (Section 4, Appendix C.3), we provide several further metrics (% resolved by repo/year/annotation label, the rates of different categories of actions issued by SWE-agent JS/M) that present resolve rates more granularly. As an immediate response, we can also include file localization (F1) scores for SWE-agent JS/M in the appendix as well.
>
> With regards to establishing new intermediate metrics, we think that such contributions might be more appropriate for methods that build upon this benchmark. For instance, with SWE-bench, different approaches (e.g. procedural vs. agentic) tend to use different signals as a way of measuring progress, and these signals are not necessarily transferable across methods. While one method might rely on test-reproduction frequently (e.g. SWE-agent), another does not use it at all (e.g. Agentless), so the utility of measuring issue reproduction success rates is dependent on the method. Also, as SWE-bench Multimodal intentionally inherits the strong characteristics of SWE-bench, we do not invest as much in new metrics in order to better illustrate that the benchmarks have the same interface for evaluation, making it easy for practitioners to develop on both simultaneously.

---

> ### Author Response · Authors · 2024-12-03
> **Response to Reviewer yGjW (Questions)**
>
> > Q1: Have the authors tried applying SWE-agent JS or SWE-agent M on the original SWE-bench tasks?
>
> We have not tried this. SWE-agent JS/M’s Agent Computer Interface (ACI) is generally a superset of the original capabilities (e.g. tools, prompts, history processing) of SWE-agent. We could run this analysis, but given that all of SWE-agent’s original capabilities are maintained in SWE-agent JS/M, we’d expect performance on the original SWE-bench task to be very similar.
>
> > Q2: How are human annotators chosen?
>
> Thanks for pointing this out, we will provide more details in the updated version. The human annotators were 10 of the authors. For easier annotation tasks (e.g. D.1 image classification), the per-image labeling was divided evenly among all annotators. For harder annotation tasks (e.g. D.4), 3/10 annotators were randomly assigned to each task instance, and the final label was assigned as majority vote.

---

### Official Review · Reviewer_RMn2 · 2024-11-04

**Soundness:** 2
**Presentation:** 3
**Contribution:** 2
**Rating:** 3
**Confidence:** 5

**Summary:**

This paper introduces SWE-bench Multimodal (SWE-bench M), a novel benchmark designed to evaluate coding agents' ability to handle real-world software engineering tasks involving visual elements. The benchmark comprises 619 task instances collected from 17 JavaScript repositories, focusing on user-facing applications such as UI design, data visualization, and interactive mapping. Through rigorous analysis, the authors demonstrate that 83.5% of these tasks require images for proper resolution, highlighting the crucial role of visual elements in software development. The study reveals significant challenges in generalizing existing software engineering systems to handle both JavaScript and multimodal contexts, with even the best-performing systems achieving only a 12.2% success rate. The paper makes several key contributions: it provides comprehensive analysis of visual content in software development issues, introduces new evaluation methodologies for assessing visual reasoning capabilities, and reveals important insights about the limitations of current systems designed primarily for Python codebases. The authors also demonstrate that more flexible, interaction-based approaches perform better than rigid, pipeline-based systems when handling multimodal tasks, suggesting important directions for future development of software engineering systems.

**Strengths:**

This work exhibits rigorous data collection and validation processes, including comprehensive human validation of task instances, careful consideration of image necessity, and thorough experimental design with detailed ablation studies. The work addresses a significant gap in existing benchmarks by incorporating multimodality and tackling real-world challenges in software development. The dataset shows a diverse range of visual content types, and well-documented collection processes with careful consideration of licensing and reproducibility.

**Weaknesses:**

The paper has several areas requiring improvement, primarily in its framing and presentation. A significant issue is that the paper sometimes misframes its contribution by emphasizing Python-centric limitations of existing benchmarks, when its true novelty lies in targeting front-end development's inherent multimodality. The motivation for choosing JavaScript repositories should be more prominently centered on JavaScript's dominant role in front-end development and its natural incorporation of visual elements, rather than being positioned as addressing a gap in language coverage.

In addition, the evaluation methodology raises significant concerns about fairness and thoroughness in baseline adaptations. A major weakness is the oversimplified approach of merely swapping Python AST parsers with JavaScript parsers when adapting existing systems like Agentless and Moatless Tools. This superficial adaptation strategy leads to potentially premature conclusions about these systems' inability to generalize to multimodal tasks. The paper lacks a convincing demonstration that sufficient engineering effort was invested in properly adapting these baselines for multimodal scenarios.

The authors should have:
- Made a more rigorous attempt to adapt these systems for multimodal input handling, beyond just language parsing
- Documented failed adaptation attempts in detail to substantiate claims about generalization difficulties
- Explored architectural modifications that could enable these systems to better handle visual elements
- Referenced and learned from existing frameworks like HyperAgent[1] that successfully handle multiple programming languages and SE tasks

Without demonstrating that serious engineering efforts were made to strengthen these baselines, the paper's conclusions about the systems' generalization capabilities are not well-supported. The low performance of adapted baselines (3.9% for Agentless vs 11.5% for SWE-agent) might reflect inadequate adaptation efforts rather than inherent limitations of these approaches. A more compelling evaluation would require investing significant engineering effort to properly adapt these systems for multimodal tasks, even if this requires substantial work. This investment is necessary to make meaningful claims about generalization capabilities and to truly understand the challenges of adapting existing systems for multimodal software engineering tasks.

Finally, Table 1's comparison between SWE-bench M and other repository-level coding benchmarks is incomplete, notably missing comparisons with important related works such as DevEval[2], CrossCodeEval[3], and RepoExec[4]

[1] HyperAgent: Generalist Software Engineering Agents to Solve Coding Tasks at Scale, https://arxiv.org/abs/2409.16299

[2] DevEval: A Manually-Annotated Code Generation Benchmark Aligned with Real-World Code Repositories,
https://arxiv.org/abs/2405.19856

[3] CrossCodeEval: A Diverse and Multilingual Benchmark for Cross-File Code Completion, https://arxiv.org/abs/2310.11248

[4] On the Impacts of Contexts on Repository-Level Code Generation, https://arxiv.org/abs/2406.11927v3

**Questions:**

1) Could you provide more details about the engineering efforts made to adapt existing systems (Agentless, Moatless) for multimodal tasks?
2) Have you considered expanding to other visually-rich domains beyond web development (e.g., game development, mobile apps)?
3) Do you plan to expand the benchmark to include other front-end frameworks or visual-heavy domains?
4) How do you envision SWE-bench M evolving as models become more capable at handling multimodal inputs?

---

> ### Author Response · Authors · 2024-12-03
>
> We thank the reviewer for the detailed and supportive comments. We restate your main points and address each of your concerns below.
>
> > W1: SWE-bench M should emphasize the visual components of JS to front-end development rather than the limitations of Python.
>
> We appreciate your insight here and on realizing the value of SWE-bench M in evaluating multimodal systems. We think that our core contributions are not necessarily limited to expanding SWE-bench to visual programming problems, but the introduction of a second major programming language family is a major contribution in itself. Especially since we believe that a critical failure point of the evaluated baselines is not directly related to their image processing abilities, but their dependence of Python-specific attributes. We believe our results, analysis and discussion provide substantial insight into the visual components of our evaluation, but we’ve additionally updated our introduction and some language to further highlight the visual testing components of SWE-bench M.
>
>
> > W2:  Insufficient effort made to adapt baselines - cannot make claims of generalization.
>
> Your response highlights an important aspect of our contribution that we would like to reiterate. In particular, our strongest claims about generalizability refer to particular systems, not necessarily to methods. However, based on our results, we also suggest to readers an important finding, which is that highly structured systems, like Agentless, may be prone to language-specific design choices that make them inherently less generalizable. Our approach deliberately preserved the core architectural assumptions of baseline systems while making only necessary adjustments, allowing us to examine their inherent capacity for generalization. While you’re correct that, perhaps, an extensively re-engineered adaptation of systems like Agentless and Moatless could provide stronger performance on SWE-bench M, the very necessity of very major adaptations demonstrates the lack of flexibility of these systems as proposed. Our experiments show that lighter weight, agent-centered systems, such as SWE-agent, simply work out of the box with essentially no language specific adaptation; where even language-adapted variants of Agentless and Moatless performed poorly.
>
> > Q1: Can we provide more details on adaptations made for Agentless / moatless?
>
> Our approach to system adaptation was guided by a principle of minimal intervention - making only those changes necessary to enable the intended functionality while preserving core system design. For Agentless, this meant implementing JavaScript-specific equivalents to their Python-specific parsers and tools, while maintaining the system's built-in assumptions about code structure and modification.
>
> > Q2: Have we considered other visually-rich domains, like game dev and mobile apps?
>
> Because the construction of SWE-bench style benchmarks is extremely expertise and labor intensive, we take a similar approach to SWE-bench when constructing SWE-bench M. This means focusing on repositories that can provide the greatest number of problems with the natural SWE-bench Issue-PR-Test structure. The resulting distribution of tasks and problems are the natural result of this task acquisition process. While other visually-rich domains such as game dev and mobile apps could further increase the diversity of problems to evaluate, we’ll leave
> the acquisition of new datasets to future work.
>
> > Q3: How do we envision SWE-bench M evolving as multimodal models improve?
>
> We see the contribution of SWE-bench M as helping guide the current and next generations of software engineering agents and LMs to focus on a greater diversity of problems, including multiple languages, front-end, and visual programming. We think that the future of the SWE and multimodal evaluations will require expanding to more fundamentally difficult problems and domains, such as long-playing game development, which could require very different solutions and models. We see SWE-bench M as being a major contribution to provide meaningful evaluation and comparison of future systems. The current performance of systems of SWE-bench M suggests that there is still a lot of improvement for these systems to make before tackling these harder domains.
>
> ---
> Thank you again for the thoughtful review and suggestions. We hope we’ve thoroughly addressed your concerns.

---

### Official Review · Reviewer_wvpw · 2024-11-06

**Soundness:** 3
**Presentation:** 3
**Contribution:** 2
**Rating:** 6
**Confidence:** 4

**Summary:**

This work introduces SWE-Bench MultiModal, an extended version of SWE-Bench, which addresses two limitations of the original version: (1) its focus solely on the Python programming language, and (2) its reliance on a single textual modality. The proposed benchmark overcomes these shortcomings by incorporating JavaScript as the primary programming language and adding images that reflect the issues being addressed, in order to assess the capability of language models (LMs) in generating multimodal patches. Additionally, the authors test the performance of existing LMs after adapting this benchmark to three systems: SWE-agent, Agentless, and RAG. Experimental results demonstrate that this benchmark remains a challenging one.

**Strengths:**

Overall, this work is commendable, and I believe the following points are particularly praiseworthy:

1. The introduction of multimodal autonomous systems for software engineering is a forward-thinking direction. By introducing this benchmark, the authors contribute to the community's understanding and research in this emerging field.
2. The experimental comparison of existing models is thorough, yet the primary aim is to highlight that this is a challenging and unsolved problem.
3. Significant contributions have been made in data collection and system adaptation, providing a solid foundation for guiding future research in this field.

**Weaknesses:**

However, from my perspective, I believe the paper could be improved in the following areas:

1. The authors claim that SWE-Bench only uses Python, limiting the ability to assess the generalization performance of language models, and use this as the motivation for proposing SWE-Bench MultiModal. However, this argument seems unconvincing. Although SWE-Bench M introduces JavaScript (and potentially other languages like HTML in the repository), a truly "generalizable" benchmark should encompass multiple programming languages, not just extend the original benchmark with a new one.

2. Despite the extensive engineering adaptations made in this work, the benchmark seems somewhat lacking in innovation compared to SWE-Bench. SWE-Bench was created from scratch (0 -> 1), whereas SWE-Bench M is an extension (1 -> 1.5, maybe).

3. Multimodal software engineering is a highly attractive topic, but it seems unlikely that, in the foreseeable future, we will be able to solve it comprehensively in both practical applications and research. As a benchmark, simply increasing the difficulty of the tasks does not seem like a sustainable or effective direction.

**Questions:**

1. I performed a preliminary review of the benchmark provided by the authors and noticed that a significant number of patches and test patches are empty. Does this suggest potential concerns regarding the quality of the benchmark?

2. I have questions about how the benchmark is evaluated. For example, how is the effectiveness of the generated patches in fixing new front-end pages assessed? The authors mentioned pixel-level visual testing, is this pixel-level comparison reasonable?

3. The authors should consider providing a complete repair process and examples to demonstrate how the system performs on the benchmark. This would offer a clearer understanding of how the system operates in practice.

4. The authors should consider showcasing several successful and failed cases to demonstrate that the successful cases across all models are not merely fixes for very simple problems. Additionally, it would be valuable to provide insights into the reasons behind the repair failures. This would help clarify the challenges and limitations of the current approach, offering a more comprehensive understanding of the model's capabilities and shortcomings.

---

> ### Author Response · Authors · 2024-12-03
>
> Thank you so much for your insightful comments and feedback. Below we summarize your main points and try to address each of them in turn.
>
> > W1: Does measuring generalization require extending to more languages than just javascript?
>
> This is a great insight, and we certainly share your sentiment on the benefit of increasing the diversity and number of languages LMs / agents are evaluated on. However, we believe that a greater number of languages is not currently necessary for evaluating and encouraging more general AI programming systems. For instance, even though our benchmark emphasizes Javascript, Typescript, and related languages, we show that existing systems already fail to generalize to this domain. As such, this shows that evaluating systems on SWE-bench and SWE-bench M already will put significant pressure on researchers to develop systems that are more language agnostic and thus more likely to generalize to different domains, languages, and paradigms. Ultimately, while we agree that future evaluations can benefit from more languages diversity, we think that SWE-bench M, as is, makes the first, and most significant, step in that direction, and can have great evaluation power; especially when combined with the original SWE-bench.
>
> > W2: Is SWE-bench M is a meaningful innovation?
>
> While SWE-Bench provides a strong foundation, SWE-Bench M addresses a critical gap between SWE-Bench and real-world software engineering challenges by introducing new language and multimodal settings. With SWE-bench M we show that systems can easily overfit to SWE-bench, and Python in particular, which makes evaluation of these systems for real world scenarios harder. We demonstrate that evaluating on SWE-bench and SWE-bench M together is a more effective evaluation, that tests approaches that are more transferable to the real-world than Python-tailored systems.
>
> > W3: Is multimodal too hard? Is increasing difficulty an effective evaluation strategy?
>
> While we agree that existing systems may not be immediately effective on SWE-bench M, we believe that this very challenge makes SWE-bench M essential for several reasons:
> First, the integration of visual and textual understanding reflects the inherently multimodal nature of software engineering in practice. Our analysis reveals that 83.5% of image-containing issues require visual comprehension for resolution - these aren't artificial difficulties we've introduced, but rather authentic challenges that real developers navigate daily. Besides filtering for issues with images, we perform no adversarial or other filtering to increase the difficulty of these problems. To shy away from such tasks would be to mischaracterize the nature of software engineering and provide a less than realistic evaluation of evaluated AI systems.
> Second, while current performance is indeed modest (12.2% success rate), this should not be interpreted as a failure of SWE-bench M, but rather as a more accurate reflection of the current capabilities and limitations of our systems. Just as early NLP benchmarks helped drive progress despite initially low performance, we believe SWE-bench M serves a crucial role in highlighting the gap between current AI capabilities and the full spectrum of real-world software engineering challenges.
>
> > Q1: Why are many patches and test patches empty in the authors' benchmark, and does this affect benchmark quality?
>
> At this time, we’ve intentionally avoided releasing full solutions and tests for the test split of SWE-bench M in order to more strongly discourage overfitting to the test split. Of course, we’ve released these details for the development split of SWE-bench M for active system development. In collaboration with our sponsors, SWE-bench M will provide an evaluation API for submitting and reporting performance on the test split.
>
> > Q2: Is pixel-level visual testing a valid way to evaluate front-end patches' effectiveness?
>
> Our pixel level testing primarily applies to instances from Chart.js, a library for chart generation that typically involves generating data visualization charts such as bar plots, scatter plots, etc., and openlayers, a library for geographic visualization and embedding maps in webpages. Visual testing component then apply to pixel-level comparisons for these types of charts and maps, which have low variation and ambiguity.
>
> > Q3: Can you demonstrate successful and failed repair cases to prove system can handle complex fixes and explain failure reasons?
>
> Based on your suggestion, we’re adding 2 failed and successful dev instance trajectories from SWE-agent to better demonstrate the nature of these problems and agent solutions. These are shown in the updated appendix.
>
> > Q4: Are we missing DevEval, CrossCodeEval, RepoExec?
>
> Based on your suggestion, we’re also updating Table 1 to include the additional benchmarks mentioned.

---

> > ### Comment · Reviewer_wvpw · 2024-12-03
> >
> > I appreciate the authors' responses to our questions one by one. Most of the responses are very reasonable, and I am grateful for that. However, my primary concern remains the completeness of the benchmark. Although the authors have stated that an evaluation API will be provided to report the performance on the test split, this work should be considered incomplete until the API's availability is demonstrated, which is a critical aspect of the evaluation. I will maintain my score until I see the API is genuinely available.

---

### Meta-Review · Area_Chair_HnRK · 2024-12-22

**Metareview:**

Summary of the paper: The paper presents SWE-Bench Multimodal, an enhanced benchmark designed to evaluate the performance of coding agents in real-world software engineering tasks that involve visual elements. Addressing limitations of prior benchmarks that focused solely on Python, SWE-bench M incorporates JavaScript as the primary programming language and includes over 600 task instances with visual components, such as screenshots and diagrams, gathered from 17 user-facing application repositories. 83.5% of these tasks require images for effective resolution, underscoring the importance of visual reasoning in software development. Despite testing existing LLMs adapted to this new benchmark, the results reveal significant challenges, with the best-performing systems achieving only a 12.2% success rate. The study also emphasizes that many current systems are constrained by Python-centric designs, which limits their applicability in multimodal contexts. Findings suggest that more flexible, interaction-based approaches are more effective than rigid, pipeline-based systems for handling these complex tasks.

Strengths of the paper:
- Addressing Real-World Challenges: Fills a significant gap in benchmarks by incorporating multimodality and reflecting real-world software development challenges, with a dataset featuring diverse visual content types.
- Robust Data Collection and Validation: Employs rigorous data collection and validation processes, including human validation, careful consideration of image necessity, and extensive manual efforts to ensure benchmark usability.
- Thorough Experimental Comparison: Provides a comprehensive evaluation of existing models, emphasizing the complexity of the problem being addressed.

Weaknesses of the paper:
- Limited Language Diversity: SWE-Bench MultiModal introduces only JavaScript alongside Python, failing to encompass a broader range of programming languages needed for a truly generalizable benchmark.
- Lack of Innovation: The benchmark appears to be a minor extension of SWE-Bench rather than a significant advancement, lacking the innovative foundation seen in the original benchmark.
- Insufficient Test Quality Enhancements: While some "flaky" tests were filtered, the overall number of Fail to Pass tests remains low, indicating a need for more rigorous test quality improvements and potential test augmentation.
- Narrow Evaluation Metrics: The reliance on binary pass/fail metrics overlooks the complexities of real-world software development, suggesting a need for additional intermediate metrics to provide deeper insights into the problem-solving process.
- Oversimplified Baseline Adaptation: The evaluation methodology's reliance on merely swapping Python AST parsers for JavaScript parsers raises concerns about the thoroughness of adaptations and the validity of conclusions regarding the systems' generalization capabilities.

Reason for the decision: During the rebuttal, the authors addressed each concern raised by the reviewers. Most of these concerns were conceptual, and the authors clarified their positions without the need for additional experiments. I find it challenging to reach a decision regarding the paper due to the divergence in opinions among the reviewers. Two reviewers recommend a weak accept, while the other two advocate for rejection, with one strongly opposing the paper. On one hand, SWE-Bench M is an incremental extension of SWE-bench that primarily introduces JavaScript with multimodal inputs. On the other hand, SWE-bench M provides valuable insights into the limitations of existing software engineering LLMs and serves as a solid resource for assessing visual reasoning capabilities in coding tasks. Given that multimodality is increasingly important—not only in software engineering but across various fields—and considering the emergence of more powerful LLMs with advanced coding capabilities (like OpenAI's o3), I slightly leaning to accept this paper because this benchmark may offer significant contributions to advancing LLM research.

**Additional Comments On Reviewer Discussion:**

As discussed above, most of these concerns were conceptual, and the authors clarified their positions without the need for additional experiments. The authors did response to each concern raised by the reviewers, but none of the reviewers changed their ratings.

---

### Decision · Program_Chairs · 2025-01-22

Accept (Poster)